# Large-scale drug sensitivity, gene dependency, and proteogenomic analyses of telomere maintenance mechanisms in cancer cells

Yangxiu Wu [1,2], Zhaoxiang Cai [2], Dale Cross[1], Jane R. Noble[1], Kelsy Prest[1], Jamie Littleboy[1], Scott B. Cohen[3], Baylee Edlundh[1], Jennifer M. S. Koh[2], Ran Xu[1], Zainab Noor [2], Milad Bastami [4], Sara Valentini[5], Laura Richardson [5], Syd Barthorpe[5], Nader Arymanesh[4], Phillip J. Robinson [2], Peter G. Hains [2], Mathew J. Garnett [5], Qing Zhong [2] ✉, Roger R. Reddel [1,2] ✉ & Karen L. MacKenzie [1] ✉

Replicative immortality is a hallmark of cancer, driven by the activation of telomere maintenance mechanisms, that is yet to be therapeutically exploited. To expedite discoveries that will enable the development of therapeutics that target telomere maintenance mechanisms, this study provides a resource of telomere biology metrics for a pan-cancer panel of 976 cell lines. We generate proteomic data from data-independent-acquisition mass spectrometry for most of these cell lines and integrate pre-existing multi-omic, drug sensitivity, and molecular dependency data from CRISPR/Cas9 knock-out screens. The data illustrate a broad range and heterogeneity in telomere biology, including states that diverge from the binary model of telomere maintenance activation involving either telomerase or the Alternative Lengthening of Telomeres mechanism. Using the telomere biology metrics and multi-omic data, we derive proteomic and transcriptomic predictors of Alternative Lengthening of Telomeres and telomerase activity levels. Our investigations also reveal molecular vulnerabilities associated with the Alternative Lengthening of Telomeres mechanism and drug sensitivity correlating with telomerase activity levels. These findings illustrate opportunities for leveraging this resource to realize the potential for telomere biology-directed cancer therapeutics and companion diagnostics.

Replicative immortality, a hallmark of cancer, is underpinned by telomere maintenance mechanisms (TMMs) that counteract telomere shortening[1]. Telomeres are comprised of 6 base pair DNA repeats protected by the shelterin complex, and play a crucial role in genome stability[2]. Most normal human somatic cells lack a TMM and consequently undergo telomere shortening in association with DNA replication, which ultimately limits cell division. The TMM most prevalent in human cancer is telomerase (TEL), a ribonucleoprotein enzyme composed of a reverse transcriptase subunit (TERT) that synthesizes telomeric DNA from the template region of an RNA subunit (TERC)[3]. A much smaller number of cancers maintain their telomeres by an Alternative Lengthening of Telomeres (ALT) mechanism that involves homologous recombination-dependent synthesis of DNA using telomeric DNA as a template[4,5]. However, the landscape of TMMs in human

cancer appears to be more complex than a binary choice of activation of TEL or ALT, as there is evidence that some cancers develop in the absence of ALT or TEL, while others exhibit evidence of both ALT and TEL. Cancer-derived cell lines without TMM were shown to undergo continuous telomere shortening, a phenotype referred to as Ever-Shorter Telomeres (EST) that may be enabled by a large telomere reserve[6,7]. The existence of cancers that either lack a known TMM (TMM-double negative, DN) or exhibit both ALT and TEL (TMM-double positive, DP) have profound implications for the future application of TMM-targeted cancer therapies. A more complete understanding of the landscape of TMM in cancer is therefore needed for telomere biology to be successfully exploited in cancer treatment.

The current study applies a range of biochemical, imaging, and molecular TMM analyses to generate a large-scale pan-cancer telomere biology dataset covering 976 cancer cell lines derived from 28 different tissue types and representing more than 60 cancer types. Here, we identify ALT, TMM-DN and TMM-DP cancer cell lines, including examples of TMM-DP cell lines that are mixtures of cells with ALT or TEL, and a cell line comprised of subclones with both TMMs activated. We also provide proteomic data for 940 of the cell lines, generated using our recently published Heat and Beat (HnB) preparative method[8], adding to the existing multi-omic data, gene dependency data from whole genome CRISPR/Cas9 knockout screens, and drug sensitivity data available for the majority of this cell line panel. These datasets are used in this study to derive an ALT classifier and an algorithm that predicts telomerase levels with high accuracy, to define molecular vulnerabilities in ALT cells, and to identify existing drugs with potential for application in TMM-directed therapy. These discoveries and resources provide opportunities for the prioritization and development of TMM targets and biomarkers for application in cancer precision medicine.

## Results

### TMM categories defined by telomere biology metrics

To generate a data resource for the discovery of molecular targets for therapeutics that exploit cancer telomere biology, we characterized TMM in a panel of 976 cell lines representing a wide range of cancer types from pediatric and adult patients. All except 25 of these cell lines are from the GDSC/Wellcome Sanger Institute (WSI) panel (Supplementary Data 1), and most are annotated with published multi-omic, CRISPR/Cas9 gene dependency, and drug response data accessible through Cell Models Passport and DepMap databases[9,10]. In this study, all 976 cell lines were assayed for C-circles (CCs; a hallmark of ALT activity), telomerase activity (TA) using the qTRAP assay, and telomere content (TC) by qPCR (Fig. 1A). Additional orthogonal telomere biology assays were performed on subsets of cell lines to validate the data range and to further investigate cell lines that deviated from canonical ALT and TEL-positive cancer cells.

The results from our initial CC, qTRAP, and TC screen of 976 cell lines spread in a continuum across 2-3 orders of magnitude (Fig. 1B, C). The TA and TC ranges were confirmed by a direct TA assay with no PCR amplification[11] (Fig. S1A), and terminal restriction fragment (TRF) analyses (Fig. 1D), respectively, using representative subsets of cell lines. In addition to showing an overall strong correlation with the TC qPCR results (Pearson's r = 0.85, P = 3.79E-26), TRF analysis demonstrated telomere length heterogeneity consistent with ALT in all cell lines that were CC-positive and TA-negative (CC+/TA−) (Fig. 1D, Fig. S1B). TC also correlated with telomere length estimates from telomere fluorescence-in situ-hybridization (FISH; n = 60), as well as TC determined from whole exome sequencing (WES; n = 946) and whole genome sequencing (WGS; n = 272) data (Fig. S1B)[12]. The overall reproducibility of the CC, qTRAP, and TC data was verified by strong correlations between results from cell lines independently cultured at WSI and Children's Medical Research Institute (CMRI), most of which were separately obtained from different sources (Fig. S1C–E).

Thresholds for positive results in TA and CC assays were set at 2% of the control cell lines MCF7 and DOS16, respectively, based on data from ten other control cell lines previously characterized as canonical ALT or TEL (Fig. 1B). As expected, the great majority of cell lines (910 of 976) were TA+/CC-, consistent with TEL, while a smaller proportion exhibited features of ALT (CC+/TA−) (Fig. 1B). A substantial minority of cell lines deviated from these phenotypes, with a subset of cell lines testing positive in both the CC and TA assays (CC+/TA+; double-positive; DP) and others exhibiting no detectable CCs or TA (CC-/TA-; double-negative; DN).

To investigate whether the DP cell lines were comprised of mixtures of ALT and TEL subpopulations or had activated both TEL and ALT in individual cells, subclones were established from three TMM-DP cell lines (SCH, PANC-08-13, and SNU-387) using flow cytometry to sort single cells. qTRAP and CC analyses showed that all 9 subclones derived from PANC-08-13 were CC−/TA+, while all 20 subclones from SNU-387 were CC+/TA− (Fig. 2A). These results indicate that the parental PANC-08-13 and SNU-387 cell lines were mixtures of ALT and TEL single-positive cells, with the TEL subclones in PANC-08-13, and the ALT subclones in SNU-387, having relative growth and/or fitness advantage when grown from single cells. In contrast to these TMM-DP mixed cell lines, all 19 SCH subclones exhibited the CC+/TA+ DP phenotype (Fig. 2A and S2A), consistent with simultaneous activation of ALT and TEL in individual cells.

Contrasting with TMM-DP cell lines, 108 cell lines were TMM-DN or exhibited low levels of either CC or TA (2-10% of controls) (Fig. 1B). To determine whether telomere length was maintained in these cell lines, representative DN, CC-Low/TA-, and TA-Low/CC- cell lines were cultured for at least 50 population doublings (PD) and subjected to TRF analysis every 10-15 PD. Quantitation of average telomere length showed that most DN cell lines underwent telomere loss at an average rate of at least 40 bp/PD during the culture period (Fig. 2B and S2B, Supplementary Data 2). Consistent with the EST phenotype[6,7], two DN cell lines that initially had very long telomeres (RPMI-7951 and MC-IXC) exhibited rapid telomere loss but continued proliferation (Fig. S2C). After a period of telomere shortening, other cell lines that were initially DN either ceased or slowed their proliferation (VMRC-MELG and EW-24) or continued to proliferate by upregulating TA (EW-11, EW-22, and SNU-475) (Fig. 2B, S2C-D). A subset of TA-Low/CC- cell lines also exhibited shortening, while others upregulated TA or proliferated steadily with stably low levels of TA (Fig. S2E). These results show that low levels of TA (2–10% control) were sufficient for telomere maintenance in some, but not all TA-Low cell lines.

CC-Low/TA- and CC+/TA- cell lines retained very long heterogeneous telomeres during culture, consistent with an ALT phenotype (Fig. S2B). However, temporal fluctuations in mean telomere length and CC levels were observed (Fig. 2B, S2B, and S2F). This included telomere lengthening in RH-41 and shortening followed by stabilization in the CC-Low cell lines ES5 and EN. CC levels were undetectable at times in the CC-Low/TA- cell line EN and were dramatically upregulated following a proliferative crisis in parallel with telomere lengthening in Hs-746T and HuO-3N1 (Fig. 2B and S2F). Two other cell lines (SW1417 and SW626) exhibited mean telomere length shortening and fluctuations in CCs and TA (Fig. S2G). Collectively, these results demonstrate a high level of heterogeneity in TMM parameters that extends beyond the existing binary model of cancer cells stably expressing either TA or ALT phenotypes.

To verify ALT activity, ALT-FISH and the ALT-associated PML Body (APB) assays were performed on CC+ and CC-Low cell lines, including representative DP cell lines (CC+/TA+ and CC-Low/TA+). ALT-FISH foci and APBs were detected in all tested TA- cell lines with CC levels greater than 10%, with 13/13 positive in the ALT-FISH assay and 20/20 positive in the APB assay (Fig. 2C–F, S3A, B, Supplementary Data 2). These results supported the categorization of these cell lines as ALT,

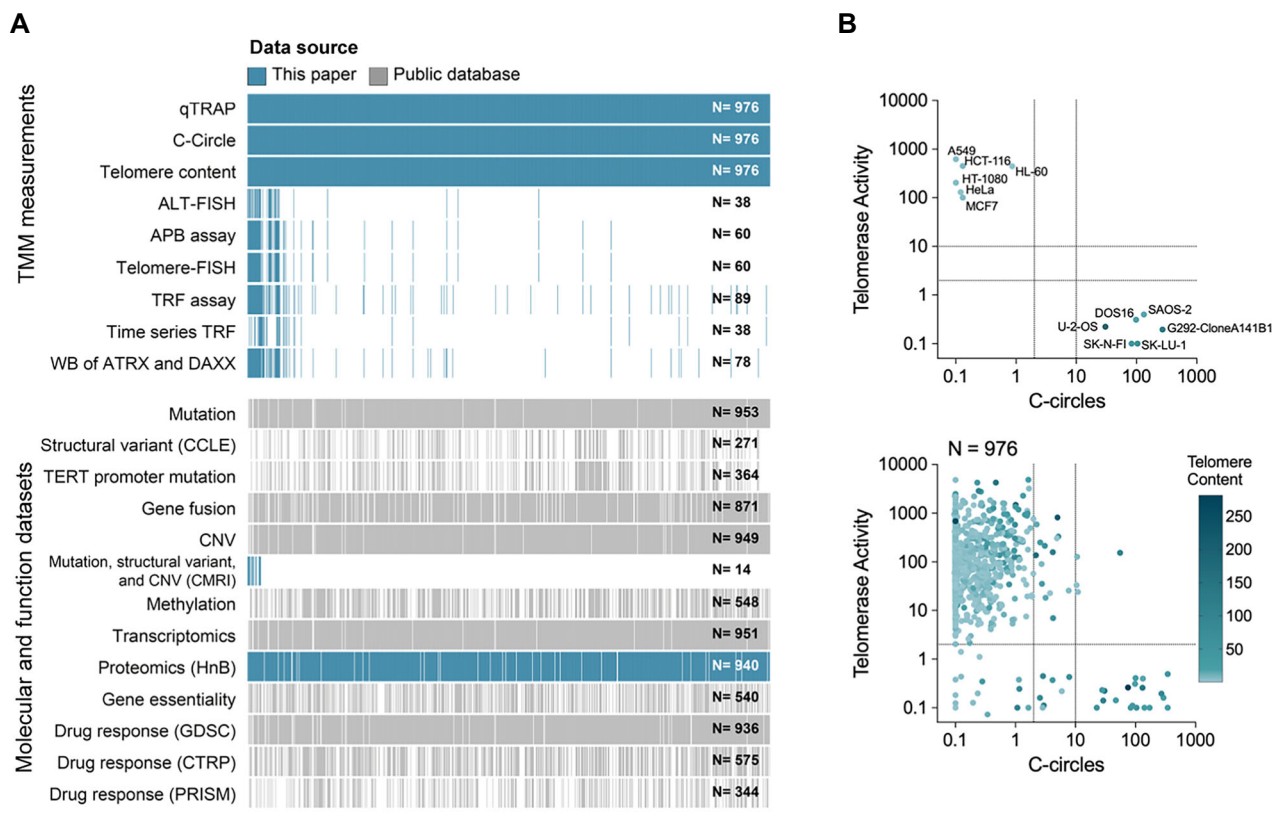

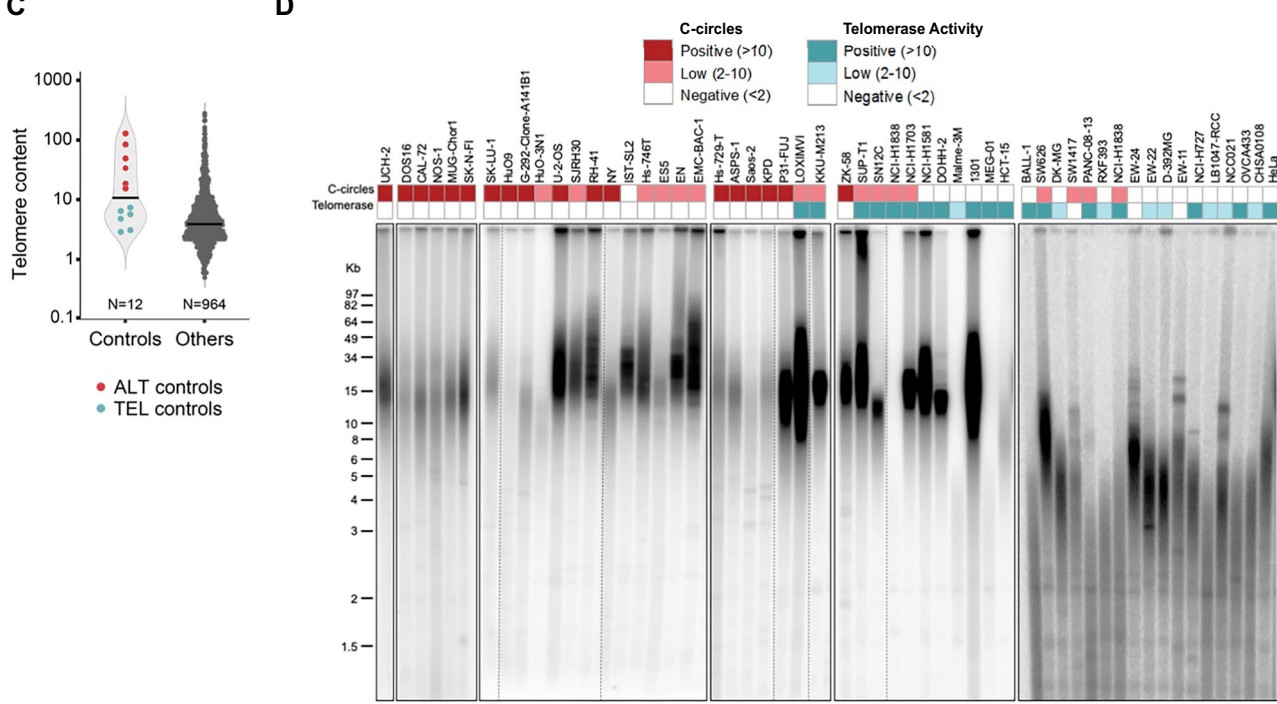

although it was noted that the tested cell lines exhibited a wider range of signal intensities and focus frequencies than the control ALT cell lines (Fig. 2E, F, S3C, D). The CC-Low/TA- cell lines had fewer foci and produced lower intensity signals than cell lines with CC > 10%, but all were positive in at least one assay, confirming moderate ALT activity. DP cell lines also produced moderate results in the ALT-FISH and APB assays, with low-intensity foci detected in a small proportion of cells. In total, 9 out of 11 DP cell lines exhibited evidence of ALT activity in either the ALT-FISH or APB assay.

TMM categories for each of the 976 cell lines were set using average values of TMM metrics (CC, TA, TC, APB, and ALT-FISH), including results from independent stocks of cell lines and multiple

**Fig. 1 | Broad range of telomere maintenance mechanism (TMM) metrics in 976 cancer cell lines. A** Data generated in this study includes results from biochemical, molecular and cell biological assays for telomere maintenance mechanisms (TMM) in 976 cell lines, additional orthogonal TMM assays on representative cell lines, proteomic data on 940 cell lines from data-independent acquisition (DIA)-mass spectrometry (MS) using heat and beat (HnB) processing methodology, Western blot (WB) and mutation/structural variant/copy number variation (CNV) analysis of ATRX and DAXX in specific cell lines. Data from public sources, accessed through Cell Models Passport, Cancer Cell Line Encyclopedia (CCLE), and the Dependency Map (DepMap), include multi-omic data, gene essentiality from CRISPR-Cas9 whole-genome screens, and drug response results. Processed methylation data were retrieved from a previous publication[112]. Vertical lines represent data on a unique cell line, and the total number of cell lines per dataset is indicated at the right of each row. ALT-FISH: Alternative lengthening of telomeres (ALT)-fluorescence in-situ hybridization (FISH); APB: ALT-associated PML bodies. TRF: Terminal

restriction fragment. CMRI: Children's Medical Research Institute. **B** Telomerase activity (TA) measured by qTRAP assay, C-circles (CCs) quantified using the CC assay with an isotope-labelled probe, and telomere content (TC) quantified by qPCR. Top panel shows results for control cell lines with well-defined TMM (N = 12). Bottom panel shows results for 976 cell lines, including controls. Dashed lines delineate negative values (less than 2%) and low positive values (2–10%) relative to control cell lines. Data are provided in the Source Data file. **C** TC in 12 control cell lines and the 964 other cell lines. Horizontal bar indicates the median. Data are provided in the Source Data file. **D** TRF Southern blots showing telomere lengths of 54 representative cell lines with varied TMM phenotypes. Dashed lines indicate where lanes were rearranged for presentation purposes, and solid lines indicate independent gels. TRF analysis was performed on a total 89 cell lines across 20 gels; most of the 89 cell lines were assayed more than once to ensure reproducibility. Uncropped gels are shown in Source Data. Mean telomere lengths calculated from TRF analysis are in Supplementary Data 2.

---

time points, where available. Based on these data, the cell lines were classified as TEL, ALT, ALT-Low, DN, DP, or Other (Supplementary Data 2, Fig. 3A, B). The ALT-Low category was defined by the absence of TA, fluctuating low-level C-circles (average 2–10% control), high TC, very long heterogeneous telomeres, and positive signals in at least one of the ALT-FISH or APB assays. The cell lines classified as "Other" included cell lines that exhibited fluctuations in TA and/or CC that traversed TMM thresholds, cell lines with limited replicative potential, and a CC-Low/TA- cell line that did not fit the ALT-Low criteria due to relatively short telomeres (MMAc). Most cell lines in the panel were categorized as TEL (93.9%), with 2.5% classified as ALT or ALT-Low and 2.9% DN or DP. ALT and ALT-Low cell lines were disproportionately represented among mesenchymal cancers (soft-tissue and bone-derived sarcomas) relative to adenocarcinomas and other cancer types. ALT cell lines were also identified in cancer subtypes for which no ALT cell lines were previously known, specifically acute myeloid leukemia, gastric carcinoma, alveolar soft-part sarcoma, and chordoma (Fig. 3B, C, Supplementary Data 2).

## Molecular and genomic variability among ALT and TEL cell lines

To investigate molecular associations with telomere biology, the TMM data was integrated with existing genomic, transcriptomic, and methylomic data (Fig. 1A). Additional WGS was performed on 14 ALT and ALT-Low cell lines, and proteomic data was generated for 940 cell lines using DIA-mass spectrometry (DIA-MS) using our recently developed HnB preparative method[8]. This approach produced a higher number of protein identifications and high replicate correlations compared with our previous proteomic study of 949 cell lines[13] (Fig. S4A–C). The current study produced quantitative data for 9039 proteins with a median of 5420 proteins per cell line.

Although ALT activation has been associated with abnormalities in the ATRX/DAXX chromatin remodeling complex[14], recent studies have shown that genomic mutations and loss of expression of *ATRX* or *DAXX* are not universal features of ALT cancer cells[15]. Here, Western blot analysis of 78 cell lines, including all ALT and ALT-Low cell lines (n = 24), showed that 11/20 ALT, 4/4 ALT-Low expressed full-length ATRX and DAXX proteins (Fig. 4A and S4D, Supplementary Data 3). Of the nine ALT cell lines that produced abnormal results for ATRX or DAXX in Western blot analysis, seven had no detectable ATRX, one expressed a truncated ATRX protein (ASPS-1), and one lacked full-length DAXX (G-292-Clone-A141B1). The ATRX western blot results were concordant with peptide detection by DIA-MS analysis (Fig. 4B and S4E). From 28 ATRX tryptic peptides detected across the cell line set, an average of less than one ATRX peptide was detected per sample replicate in the cell lines with abnormal ATRX expression by western blot analysis (Supplementary Data 3). Proteomic data for DAXX was inconclusive as only seven tryptic peptides were detected across the cell line panel, consistent with the relatively small size of this protein. Abnormal expression of ATRX or DAXX tended to be more

prevalent in the osteosarcoma cell lines (Fig. S4F) and was associated with significantly higher C-circle and APB scores (Fig. 4C, D).

WGS and WES data revealed gene rearrangements and large deletions that accounted for abnormal ATRX/DAXX expression in eight out of nine cell lines that lacked full-length protein by Western blot analysis (Fig. 4B). This included *ATRX* structural variants and copy number loss, an *ATRX* inversion in SAOS-2, and a previously reported *DAXX* fusion in G292-Clone-A141B1[16]. Single-nucleotide variants (SNVs) were detected in *ATRX* and *DAXX* in all TMM categories, including cell lines that expressed full-length ATRX and DAXX proteins. Together, the data demonstrate that altered ATRX and DAXX protein expression resulting from genomic alterations corresponds with strong phenotypic indicators in ALT cell lines but are not ubiquitous across this TMM group. No other statistically significant genomic or proteomic associations with ALT (including ALT-Low) were identified (Fisher's Exact test).

Associations between TA and 'omic data were also investigated. *TERT* mRNA expression correlated with TA measured by the qTRAP assay. However, outliers were apparent, including some TA-negative cell lines (ALT, ALT-Low, and DN) with *TERT* mRNA levels comparable to cell lines with high TA, and TEL cell lines with no detectable *TERT* mRNA (Fig. 4E). In contrast, TERC levels determined by RNAseq correlated weakly with TA, with almost half of the TA- cell lines expressing TERC above median levels (Fig. S4G).

Mechanisms associated with TEL activation during cancer development include activating mutations in the *TERT* promoter and methylation of the *TERT*-methylated oncological region (THOR)[17,18]. Our analysis of data from 367 cell lines revealed that *TERT* promoter mutations were associated with higher levels of TERT mRNA and lower levels of THOR methylation (Fig. 4F, G). However, TA levels, as determined by qTRAP, were not significantly associated with either *TERT* promoter mutations or THOR methylation levels (Fig. 4H, I). These results highlight the complex regulation of TA and the need for biochemical assays.

## Derivation of an ALT classifier and TA score from multi-'omic data

Since TMM cannot be reliably inferred from *ATRX/DAXX* and *TERT* 'omic data, we applied machine learning to derive RNA and protein-based classifiers for TMM groups and prediction of TA levels. Using protein and RNA levels from the HnB proteomic data and WSI transcriptomic data, respectively, we generated binary classifiers that distinguish ALT (including ALT-Low) from other TMMs (ALT predictors) (Fig. S5A, Supplementary Data 4). To mitigate inherent imbalance in tissue types among different TMM groups, the training model for the ALT predictor was down-sampled by restricting it to cancer types where ALT and/or ALT-Low were represented. Cell lines in the DP and Other categories were excluded because of the heterogeneity in these groups. Sequential Forward Selection (SFS) was used to identify the

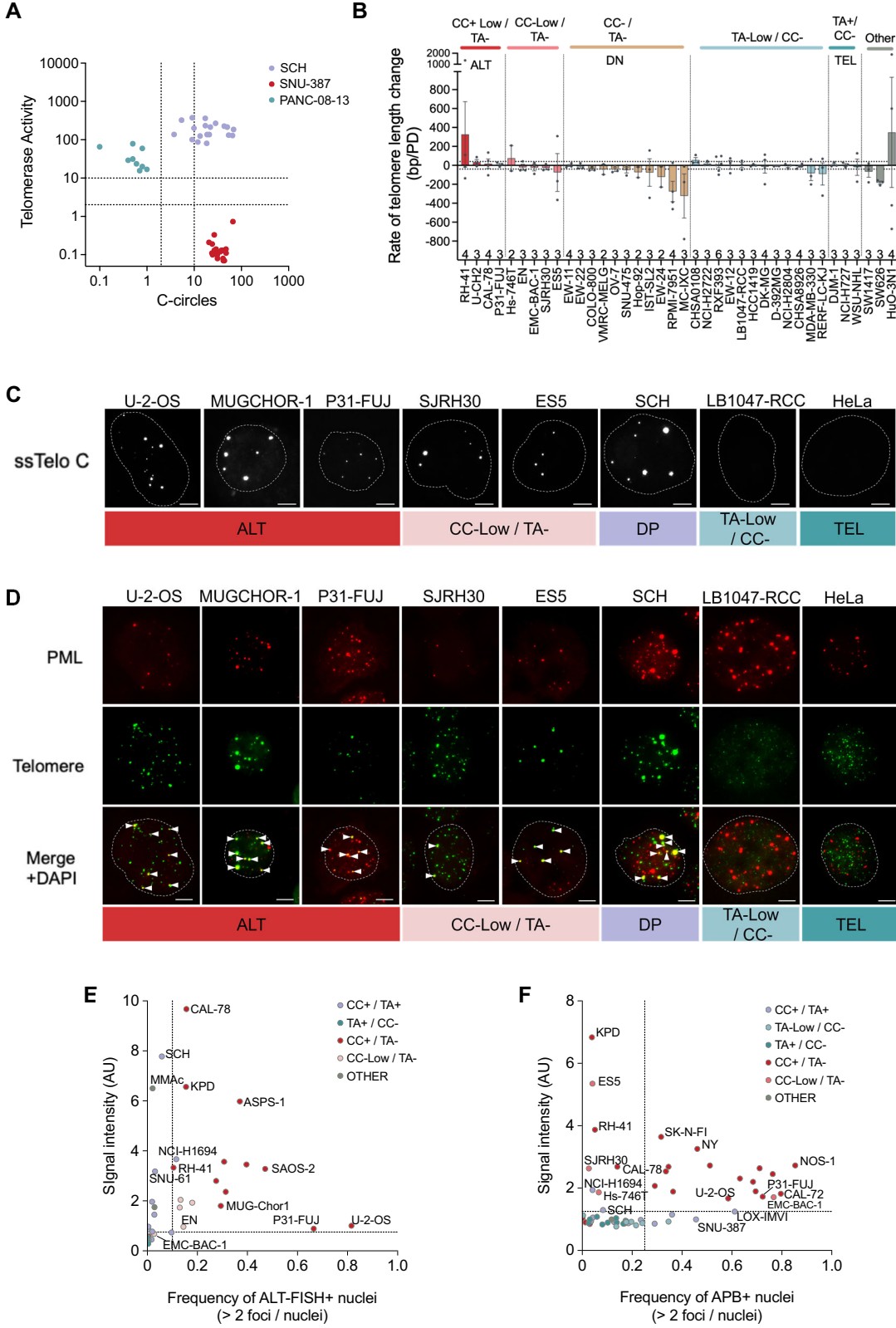

top five features from protein, RNA, and integrated protein/RNA data sets. Notably, the top RNA features for ALT prediction were TERT (negative association) and USP1 (positive association). USP1 is a deubiquitinase that may be functionally linked to ALT through its role in DNA replication and repair involving the regulation of Fanconi Anemia proteins and PCNA[19,20]. In univariate analysis, USP1 RNA was significantly higher in ALT (including ALT-Low cell lines compared with all

others ($P = 0.0011$, two-sided Wilcoxon rank sum test). TERT RNA was not expressed in most ALT cell lines (Fig. 4E). Neither TERT nor USP1 were detected by DIA-MS proteomic analysis. Using a class weight option to further mitigate the effects of imbalance between the TMM groups, the classifiers showed excellent performance in cross-validation, with the overall strongest prediction accuracy generated using the integrated protein/RNA data for prediction of ALT, achieving an

**Fig. 2 | Heterogeneity of TMM states. A** TA and CCs in subclones established from single cells sorted by flow cytometry from representative TMM-double positive (DP) cell lines. Values are means from two time points. The gating strategy used to sort single cells using flow cytometry is shown in Fig. S2A. A representative CC assay is shown in Fig. S2B, and data from biological replicates are provided in Source Data. **B** Changes in mean telomere length determined by TRF analysis of cell lines representing varied TMM phenotypes in long-term culture. Telomere length was measured in each cell line at a minimum of 3 time points, with at least 10 population doublings (PD) between each time point. Columns and error bars show the mean rate of telomere length change $\pm$ standard error of the mean (SEM) from consecutive time increments. The number of time increments is shown for each sample along the top of the x-axis. Dashed horizontal lines indicate mean telomere length change $\pm$ 40 bp/PD. PD: population doublings. **C** Detection of single-stranded C-

rich telomeric DNA by ALT-FISH assay of cell lines representing various TMM states. Dotted lines outline the nuclear membrane as defined by DAPI stain. **D** APBs indicated by arrowheads in representative cell lines. Scale bars in (**C**) and (**D**) are 5 µm. **E** Quantitation from ALT-FISH assays for 36 representative cell lines. **F** Quantitation from APB assays for 59 representative cell lines. Quantifications in (E-F) use data from at least 500 nuclei per cell line. Horizontal and vertical lines indicate midpoints between the highest values for control telomerase-positive (TEL) cell lines and the lowest value for the control ALT cell lines shown in Fig. S3C, D. Data are provided in the Source Data file. ALT: Alternative lengthening of telomeres (CC > 10% and TA < 2% controls); DP: TA > 2% and CC > 2% controls; CC-Low: CC is 2%-10% control; TA-Low: TA is 2%-10% control; DN: CC < 2% and TA < 2% controls. TEL: TA > 10% and CC < 2% controls.

area under the receiver operating characteristics (AUROC) curve of 0.99 (Fig. 5B, C, S5C, D). The RNA-based TMM predictors were further tested on a previously published external cohort of cell lines and mesenchymal stem cells ($n = 14$) and on liposarcoma tumor tissue samples ($n = 18$), which were assigned TMM classifications using TA and ALT assays[21,22]. The ALT classifier yielded highly accurate results in both cohorts, producing AUROC > 0.98 and AUROC = 0.91, respectively (Fig. 5B, C).

Quantitative predictors of TA (TA Scores) were derived by training proteomic, transcriptomic, and integrated proteomic/transcriptomic data sets on qTRAP results and applying the best five features identified by SFS in linear regression models (Fig. S5A, Supplementary Data 4). TERT RNA intensity again emerged as the top feature in both the RNA and protein/RNA models, while LAMP1, a lysosome-associated membrane protein, was selected from all input datasets (RNA, protein, and RNA/protein) (Fig. 5D). In cross-validation tests, correlations were observed between each of the five-feature TA Scores and actual TA values, with the highest correlation generated with the protein/RNA feature set (Fig. 5E, F). There is currently no published dataset with both TA and proteomic data available for testing the protein-based classifiers; however, we verified that our RNA-based TA Score surpassed TERT RNA and a previously published RNA-based 13-gene TA score (referred to as EXTEND) when tested on independent datasets[23-25] (Fig. 5E). Collectively, these results demonstrate the strong potential of 'omic'-based machine learning models derived from the TMM data generated here to predict ALT and TA levels with high accuracy.

## Proteo-transcriptomics reveal contrasting biological properties of ALT and TEL cells

To provide insight into the phenotypic properties of TMM states, we performed differential comparisons of ALT (including ALT-Low) and TEL cell lines, as well as correlation analysis of TA levels using proteomic and transcriptomic data (Fig. 6A, B, S6A, B, and Supplementary Data 5). Tissue type was applied as a covariate to account for the disproportionate representation of specific cancer types among ALT cell lines. Enrichment analysis was then performed on features differentially expressed in ALT versus TEL cell lines or correlating with TA levels (adjusted $P < 0.05$). Overall concordance was observed between the biological pathways identified by the proteomic and transcriptomic analyses. The Gene Ontology (GO) biological processes that positively correlated with ALT and negatively correlated with TA featured processes that underpin tumor progression and metastasis, such as cell motility, angiogenesis, and extracellular matrix (ECM) reorganization (Fig. 6C, Supplementary Data 5). These findings were underscored by strong correlations with the cancer hallmark pathway Epithelial-Mesenchymal Transition (EMT), which is intrinsically linked to signaling pathways that promote cell migration, invasion, and metastasis. Key modulators of EMT upregulated in ALT cells or negatively associated with TA levels include proteins involved in TGFβ signaling (TGFB1, TGFB2, TGFBR1, and SMAD3), CTNNB1, and

NOTCH2. EMT is also associated with heightened inflammation and immunosuppression, evidenced in these analyses by enrichment of pathways such as TNFα signaling via NF-κβ, IFNγ response, neutrophil-mediated immunity, IL2-STAT5 signaling, and IL6-STAT3 signaling in ALT cells and negatively correlating with TA levels (Fig. 6C, Supplementary Data 5).

Proteins with high abundance in TEL relative to ALT cell lines or showing a strong positive correlation with TA level were enriched in pathways related to cell division, cell cycle progression, DNA repair, and biological processes that support a high rate of proliferation, such as ribonucleoprotein complex biogenesis, RNA processing, and mitochondrial gene expression (Fig. 6C). These included PLK1, MKI67, replication factor C components, DNA polymerases, PCNA, and minichromosome maintenance proteins (Supplementary Data 5). A correlation between TA levels and proliferation rates was confirmed using the proliferation index calculated from CellTiter-Glo assays for 942 cell lines (Fig. S6C). Transcriptomic data also showed a significant correlation between TA level and Stemness Index, a computational prediction score of self-renewal, and differentiation potential[26] (Fig. S6D, Supplementary Data 5). Interrogation of the ChaperoneNet database[27] showed that molecular chaperones segregated according to TMM. HSP90 proteins, HSP90 co-chaperones, and subunits of the CCT/TRiC chaperonin complex were more abundant in cell lines with high TA, while small molecular chaperones, members of the cyclophilin family, and endoplasmic reticulum chaperones were upregulated in ALT cells (Fig. 6A, C, and Supplementary Data 5). These data show that activation of ALT and TA is associated with distinct proteomic and transcriptomic landscapes corresponding with divergent cancer cell phenotypes.

## Gene dependencies associated with TMM and telomere content

To investigate molecular vulnerabilities associated with TMM activation during cancer development, we analyzed data from an integrated CRISPR-Cas9 gene essentiality screen covering the majority of the cell line panel[28,29]. Dependency scores for 17,486 genes in 540 cell lines, comprised of 15 ALT (including ALT-Low) and 504 TEL cell lines, were compared in differential analysis, using tissue type as a covariate. The results identified 23 and 13 preferentially essential genes (PEGs) for ALT and TEL groups, respectively (adjusted $P < 0.05$; Fig. 7A, B and Supplementary Data 6). Over-representation analysis revealed that ALT PEGs were enriched for genes involved in DNA replication and repair, including the replication fork processing genes, SMARCAL1, FANCM, and SAMHD1, as well as two members of the alternative PCNA loading clamp, CHTF18-replication factor C complex (CHTF18 and DSCC1) (Fig. 7C and Supplementary Data 6). TERF2IP, encoding a subunit of the shelterin complex, also demonstrated essentiality in ALT compared to TEL cells. Conversely, these analyses revealed that ALT cells were comparatively tolerant to perturbation of genes involved in strand invasion processes (XRCC2 and RAD51D) (Fig. 7A, C).

For experimental validation of dependencies associated with ALT, we focused on SAMHD1, a protein with dual functions relevant to telomere biology that was highly ranked in differential comparisons and

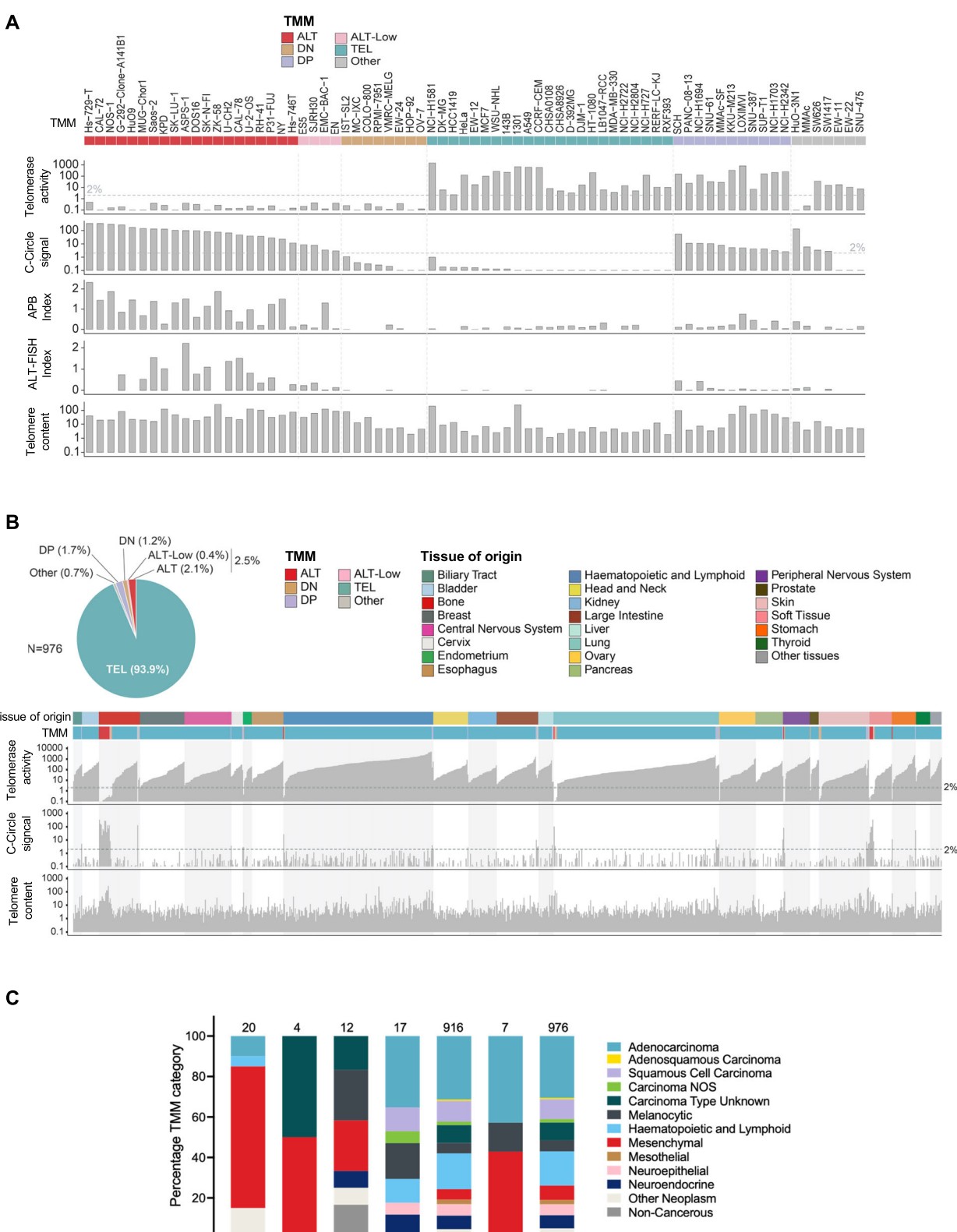

**Fig. 3 | Categorization of TMM. A** Comprehensive TMM data for 73 representative cell lines. TA and CC values were acquired in technical replicates, then means were derived from multiple time points and different cell stocks independently grown at CMRI and Wellcome Sanger Institute (WSI). **B** Representation of TMM measurements and categories across 28 tissue lineages for 976 cell lines. **C** Distribution of TMM across different cancer categories. The number of cell lines in each TMM category is shown above the columns. ALT-Low: TA < 2% control, CC average 2–10% control, high TC, very long heterogeneous telomeres, and positive signals in at least one of the ALT-FISH or APB assays. Data for Fig. 3A–C are provided in Supplementary Data 2.

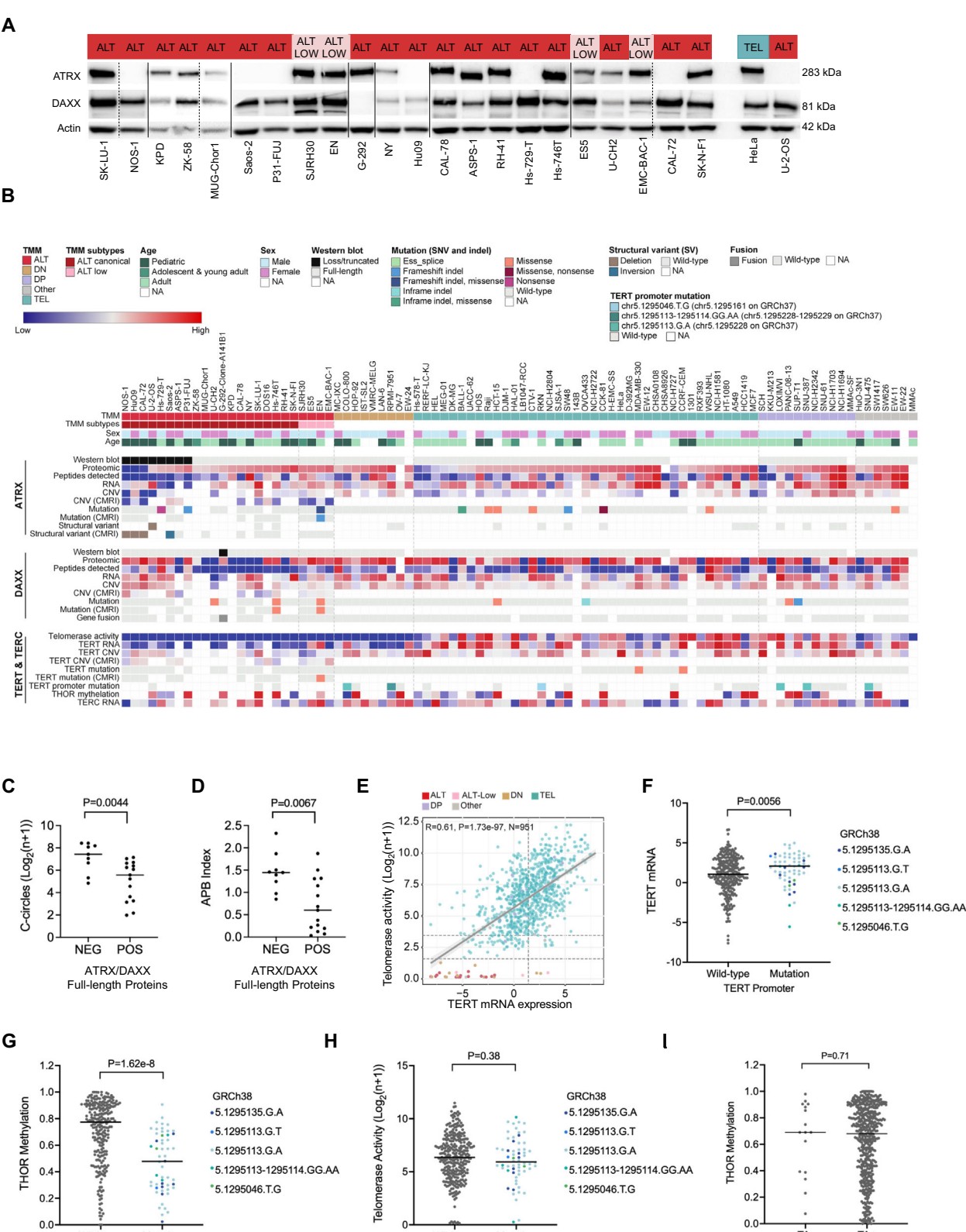

was not previously identified as an ALT vulnerability[30]. Here, *SAMHD1* expression was suppressed in a panel of eight cell lines (four ALT and four TEL) using two different siRNAs and a non-targeting control (Fig. S6A). Proliferation was monitored over seven days, and cell viability was assessed 72 hours after transfection. Consistent with the high-throughput whole-genome CRISPR screen, *SAMHD1* suppression had no impact on the proliferation of any of the four TEL cell lines. In

contrast, two ALT cell lines, CAL-72 and Hs746T, which had strong dependency scores in the CRISPR screen, exhibited impaired proliferation (Fig. S6B, C). The two other ALT cell lines, SAOS-2 and HuO9, showed no response to *SAMHD1* knockdown, which is also consistent with the CRISPR screen. The results from the viability assay were consistent with the proliferation assay and CRISPR screen data (Fig. S6C–E). Together, these results validate *SAHMD1* as an ALT-

**Fig. 4 | Molecular variability within TMM categories. A** Representative Western blot analysis of ATRX and DAXX proteins in 22 ALT and ALT-Low cell lines, plus HeLa and U-2-OS as positive and negative controls for ATRX, respectively. G-292 (G-292-cloneA141B1) serves as a negative control for DAXX expression. Actin was used as a loading control. Vertical lines distinguish independent gels, and dashed lines indicate where lanes were rearranged or excised for presentation purposes. ATRX/DAXX Western blot analyses were performed on a total of 78 cell lines across 16 independent Western blots (Fig. S4D and Supplementary Data 2). Uncropped gels for this figure are shown in Source Data. **B** Molecular profiling of *ATRX*, *DAXX*, and *TERT* in 89 cell lines representing various TMM categories. Data represented include mutations (exonic single-nucleotide variations, exonic small insertions, and deletions) and CNV. Data is provided in Supplementary Data 3. NA: Data not available. **C** CC levels and (**D**) APB Index in ALT and ALT-Low cell lines assayed for full-length ATRX and DAXX expression by Western blot, where APB Index is foci intensity x frequency of positive nuclei. *N* = 9 for ATRX/DAXX negative, *N* = 15 for ATRX/DAXX positive. **E** Two-sided Pearson's correlation analysis between TA

measured by qTRAP and TERT mRNA from RNAseq data. The lower horizontal line indicates TA = 2% control, and the upper line shows TA = 10% control. Shading indicates the 95% confidence interval. **F** TERT mRNA levels from RNAseq data (*N* = 302 for samples with *TERT* promoter wild-type, *N* = 62 for samples with a *TERT* promoter mutation), **G** Methylation of the *TERT* hypermethylated oncogenic region (THOR). (*N* = 252 for samples with *TERT* promoter wild-type, *N* = 53 for samples with a *TERT* promoter mutation), and (**H**) TA measured by qTRAP comparing cell lines with and without activating *TERT* promoter mutations (*N* = 304 for samples with *TERT* promoter wild-type, *N* = 62 for samples with a *TERT* promoter mutation). (**I**) THOR methylation in cell lines with and without expression of TA. TA +: TA > 2% control (*N* = 512), TA-: TA < 2% control (*N* = 14). *P*-values in (**C**, **D**) and (**F–I**) are from two-sided Wilcoxon rank-sum tests. Horizontal bars indicate the median. All cell lines in our panel with data available from these sources were included in the statistical comparisons. Data for *TERT* promoter mutations, THOR and TERT mRNA were from external sources[9,10,112]. Data for (**C–I**) is provided in Source Data.

associated PEG relative to TEL cells, although our data show it is not universally essential across all ALT cell lines.

Linear regression analysis showed a positive correlation between TA levels and PEGs involved in mitochondrial biology, including genes associated with the mitochondrial matrix, mitochondrial ribosomes, and mitochondrial inner membrane (7D-F, Supplementary Data 6). PEGs correlating with low TA levels reiterated results from the binary comparison of ALT versus TEL cell lines, identifying genes involved in DNA replication and repair. When ALT cell lines were excluded from these analyses, these gene dependencies were no longer significant, consistent with their specific association with the ALT mechanism rather than the lack of TA (Fig. S7A, Supplementary Data 6). PEGs that remained associated with low TA when ALT cell lines were excluded included *ACD* (encoding TPP1), a shelterin component that plays a crucial role in telomerase recruitment and processivity, and genes relating to tyrosine kinase growth factor signaling, such as *EGFR* and its downstream signaling mediators *GRB2* and *PTPN1* (Fig. S7A, Supplementary Data 6). Mitochondrial genes remained strongly associated with high TA when ALT cells were removed from the analysis (Fig. S7A, B).

Linear regression analysis of gene dependencies and TC demonstrated a negative correlation between TC and PEGs with well-established roles in telomere maintenance, namely *TERT*, the shelterin components *TERF1* and *TINF2*, and all three components of the CST complex (*CTC1*, *STN1*, and *TEN1*) (Fig. S7C, D; Supplementary Data 6). The CST complex plays a critical role in counteracting telomere loss by facilitating C-strand fill-in by Polα-primase and negatively regulates telomerase-mediated telomere extension[31,32]. *TERF2IP* had the strongest positive correlation with TC, consistent with its identification as a PEG for ALT cells, which invariably had very high TC. When ALT cells were excluded from the analysis, *TERF2IP* fell below the significance threshold, while *TERT*, *TERF1*, *TINF2*, and *CST* gene dependencies remained significantly associated with low TC (Fig. 7G, Supplementary Data 6). An additional 65 PEGs predominantly related to mitochondrial function correlated with high TC when ALT cell lines were excluded from regression analysis. Together, these results show that ALT cells have preferential dependence on DNA replication and fork processing factors, as well as *TERF2IP*, while high TA and high TC are associated with heightened dependence on mitochondrial function. Non-ALT cells with low TA exhibited stronger dependence on the telomerase processivity factors *ACD* and *POT1*.

### Drug sensitivities associated with ALT and telomerase

To identify drug sensitivities associated with TMM, we analyzed the GDSC dataset from high-throughput drug screens performed at the WSI[33,34]. In differential comparisons of Area Above the Curve (AAC) data from drug response tests of ALT and TEL cell lines, we found the FGFR inhibitor PD173074 had the highest specificity for ALT cell lines

(Fig. 8A). Although the adjusted P-value was just below our significance threshold (Adjusted *P*-value = 0.0573, *P* = 0.0014), this result is consistent with an independent study that reported the sensitivity of ALT cells to this drug[35]. Our result reflected a 2.9-fold lower average IC50 in ALT compared with TEL cell lines, with strong results in bone-derived cell lines, including osteosarcoma (Fig. S8A). We further investigated whether the sensitivity of ALT cells to PD173074 was related to expression levels or dependency on the drug's known targets. However, no statistically significant association between ALT and any measure related to FGFR genes was identified.

Among the drugs that showed differential activity toward TEL cell lines, Pevonedistat, an inhibitor of NEDD8-activating enzyme (NAE1)[36], showed a positive correlation in tissue normalized linear regression analysis with TA (Rn=0.31, Adjusted *P*-value = 3.5e-19) (Fig. 8B; Supplementary Data 7). On average, cell lines with high TA (TA > 500% control) exhibited IC50 values that were 19-fold lower than cell lines with low TA (TA = 2–10% control) (Fig. S8B). Pevonedistat activity also correlated with TA in the independent CTRP and PRISM datasets generated by the Broad Institute[37–39] (Fig. S8C; Supplementary Data 7). Tissue and cancer type analysis of the GDSC data set revealed tissue-specific variations in the strength of the correlation, with robust correlations (R ≥ 0.5) between TA and Pevonedistat activity in cell lines of the myeloid lineage, central nervous system, and pancreatic tissue (Fig. 8C, D; Supplementary Data 7).

Previous studies have shown that 'omic features can predict drug sensitivity of cancer cells[13,33,34,40]. Here, we applied an integrated dataset of TMM parameters and 'omic data for the prediction of Pevonedistat sensitivity. Using the SFS strategy (with adjustment for tissue bias) to select the best ten features from genomic mutation, transcriptomic, proteomic, and TMM data (TA levels, CC, and TC), TA was the highest-ranked single feature correlating with Pevonedistat sensitivity (Fig. 8E). When all ten features were applied together, the model produced an R-squared value of 0.48 across the pan-cancer cell panel. In cancer-specific analysis, strong results were observed for chronic myelogenous leukemia, osteosarcoma, bladder carcinoma, glioma, and thyroid carcinoma with R-squared >0.90 (Fig. 8E right panel, Supplementary Data 7).

We investigated a possible association between TA and Pevonedistat's known target pathway using multivariable regression analysis that included gene dependencies and protein expression levels of *NAE1*, *NEDD8*, the NAE1 dimerization partner UBA3, as well as the downstream mediator UBE2M (Fig. S8D, Supplementary Data 7). The results showed correlations between Pevonedistat and either TA or the TA Score to be stronger than correlations between the NAE1 pathway-related variables. It was noteworthy, however, that both Pevonedistat and TA exhibited strong correlations with the Proliferation Index in these analyses. Collectively, the results provide strong evidence that high TA is an indicator of Pevonedistat sensitivity and suggest that the

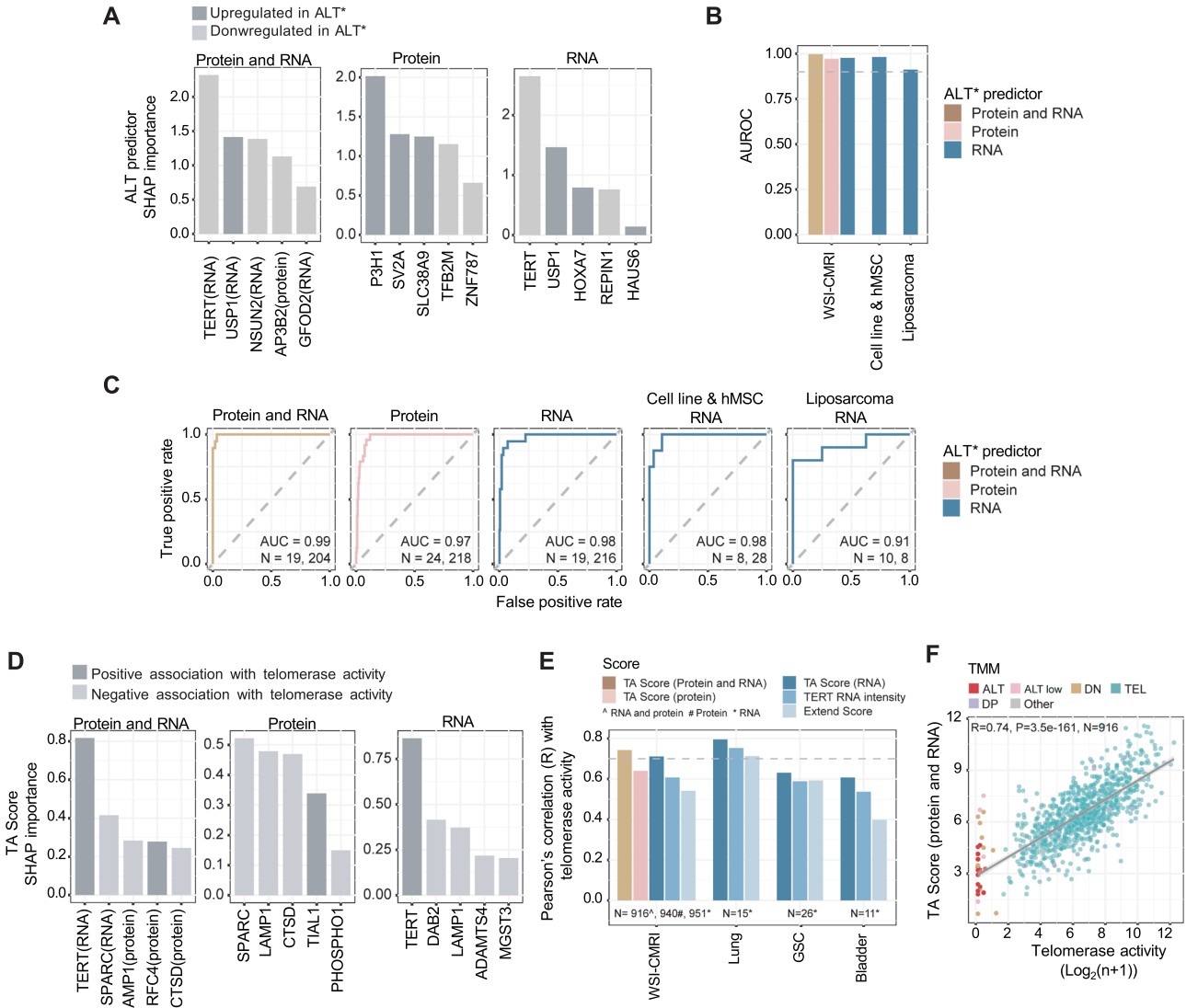

**Fig. 5 | Prediction of ALT category and TA level using proteotranscriptomic data. A** Shapley additive explanations (SHAP) values for the top five ALT-associated features selected from binary models using proteomic, transcriptomic, and integrated proteomic and transcriptomic data sets. Each feature was independently identified by sequential forward selection (SFS). **B** Area under the receiver operating characteristic (AUROC) curves of ALT predictive algorithms using the five features selected in (**A**) as input, showing 5-fold cross-validation prediction on our samples (WSI-CMRI) and independent validation on external cohorts. Only cell lines from cancer types with ALT representation were included, and the number of cell lines (N) was also dependent on the availability of the 'omic data. The dashed line indicates AUROC = 0.90. **C** Receiver operating characteristic (ROC) curves for ALT predictors from (**A**, **B**). N indicates the number

of cell lines in each group. **D** SHAP values for the top five features associated with TA levels. Each feature was independently derived by SFS of proteomic, transcriptomic, and integrated proteomic/transcriptomic datasets. **E** Two-sided Pearson's correlation between TA measured by qTRAP and TA Scores from (**D**) when tested internally in 5-fold cross-validation prediction analysis (WSI-CMRI) and against independent cohorts with published TA measurements[23,24]. Lung: lung cancer cell lines; GSC: glioma sphere-forming cells; Bladder: bladder cancer cell lines. Also shown for comparison is TA correlation with TERT RNA intensity (alone) and the previously published EXTEND Score[23]. N: number of cell lines analyzed. **F** Correlation between TA measured by qTRAP assay and values predicted by our RNA plus protein-based TA Score. Shading indicates the 95% confidence interval. *ALT includes ALT-Low. Detailed results are provided in Supplementary Data 4.

---

association between Pevonedistat and TA may involve an off-target pathway linked to cell proliferation.

## Discussion

The potential for exploiting TMMs in the development of new cancer treatments is widely recognized but largely unrealized. This study provides a resource of TMM data on a pan-cancer set of 976 cell lines to substantially increase the number of cancer models for discovering drug targets and companion biomarkers related to telomere biology. The data add a unique layer to the multi-omic, gene dependency, and drug response annotations available through the Cell Models Passport, DepMap, and other resources. Integrating TMM and proteomic data with these data sets, this study revealed phenotypic traits and

biological processes associated with ALT and TA, enabled the derivation of predictive classifiers, and uncovered TMM-associated molecular and therapeutic vulnerabilities.

ALT cell lines identified here broaden the scope for studying ALT to cancer types that include acute myeloid leukemia, alveolar soft-part sarcoma, gastric carcinoma, and chordoma. The overall representation of ALT in this cell line panel was 2.5%, lower than the 3.6% frequency reported in a large pan-cancer survey, and considerably less than the 10–15% widely suggested in the literature[41]. ALT cell lines may be underrepresented in this panel and more broadly among cell lines because they tend to be slow-growing and may be less likely to become established in standard culture conditions than TEL cells. Conversely, ALT may have been overestimated in studies where it is assumed that

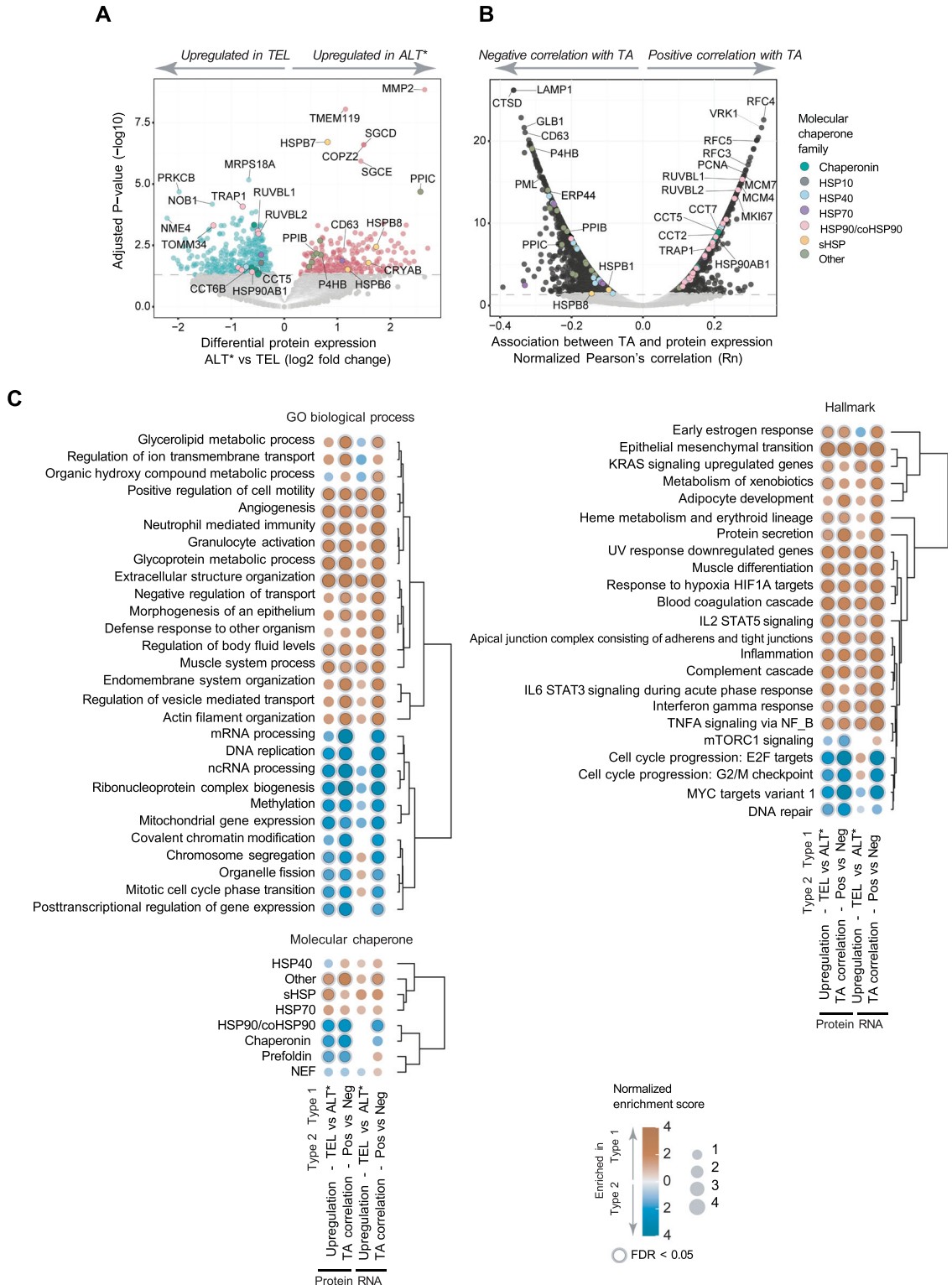

**Fig. 6 | Proteomic analysis reveals divergent cancer cell properties associated with ALT and telomerase activation. A** Differential comparison of protein expression in ALT (including ALT-Low) versus TEL cell lines. Tissue type was used as a covariate (*N* = 906). **B** Normalized two-sided Pearson's correlation (Rn) analysis of protein expression relative to TA using proteomic data acquired by DIA-MS. Dashed line in (**A**, **B**) indicates adjusted *P*-value = 0.05, calculated using the Benjamini-Hochberg (BH) method applied to *P*-values from two-sided tests (*N* = 940 cell lines). **C** Gene set enrichment analysis of proteins from differential and correlation analyses in (**A**) and (**B**) and genes from RNA analysis shown in Fig. S5B, C. Results are shown for redundancy-reduced terms (by weighted set cover method) with False Discovery Rate (FDR) < 0.05 in at least one group. *ALT includes ALT-Low. Detailed results are provided in Supplementary Data 5.

any tumor that is TA- must be ALT, not taking into account DN cancers and/or the possibility of false-negative results caused by inhibitors of *Taq* polymerase that confound qTRAP analysis of tumor tissue[42]. Because the cell line panel utilized in this study was based on the availability of multi-omic and other external datasets, it is not balanced for tissue or cancer types across TMM categories. This necessitated the implementation of various strategies to control for tissue-specific effects in downstream analysis. Notwithstanding this limitation,

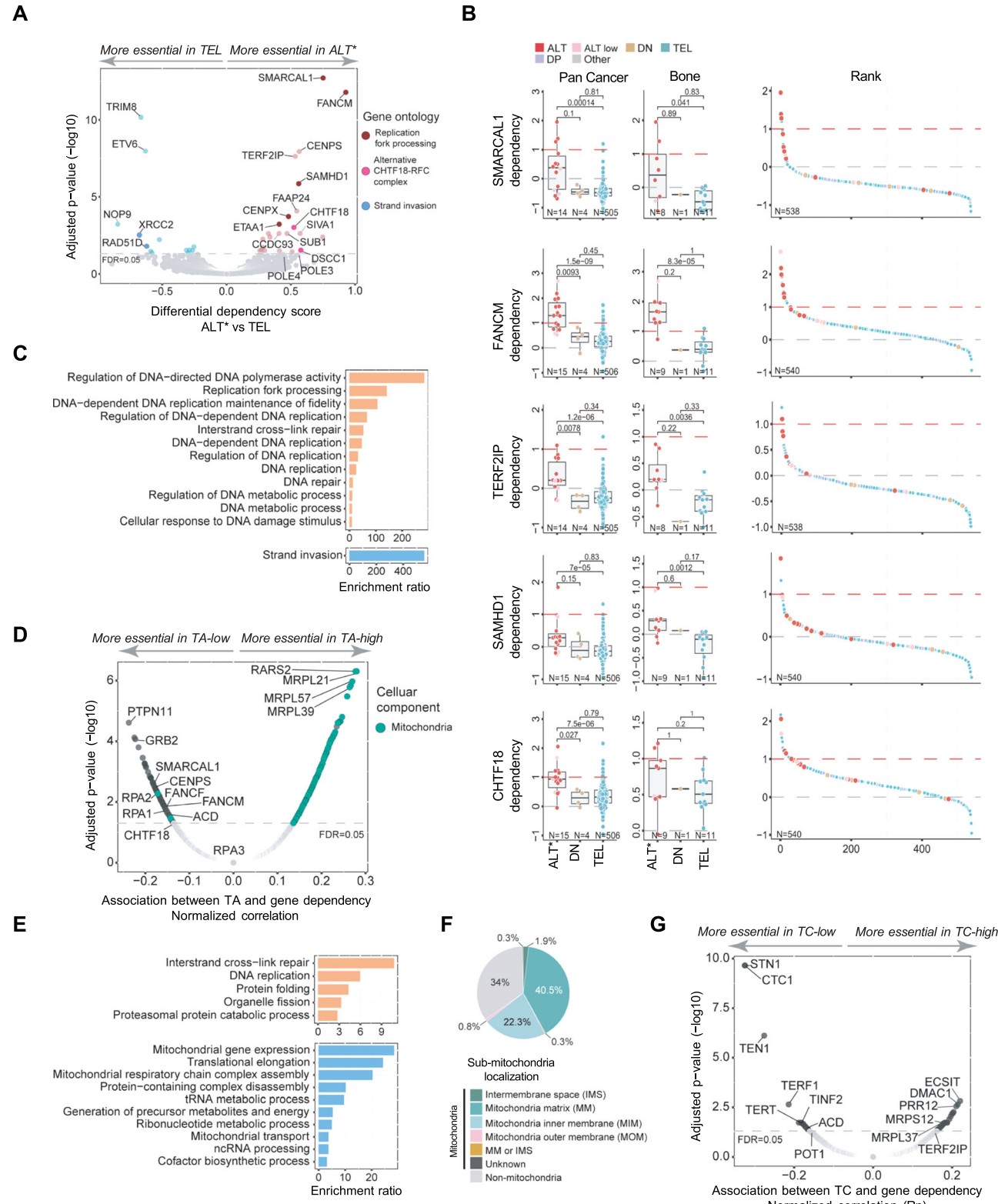

proteomic and transcriptomic TMM classifiers, such as those developed in this study, provide opportunities to identify ALT in cancer sample cohorts.

This study reveals a high degree of phenotypic heterogeneity among the ALT cell lines, evidenced by a broad range in CC levels, varied frequency and signal intensity of APBs, and disparate *ATRX/DAXX* status. More than half of the ALT cell lines expressed full-length ATRX and DAXX proteins. Cell lines that lacked full-length ATRX or DAXX had higher CC levels and APB scores than ALT cells that

expressed ATRX and DAXX as full-length proteins. This may contribute to bias toward the detection of ALT cancers with abnormal ATRX or DAXX. All four cell lines designated as ALT-Low expressed full-length ATRX and DAXX proteins, but were consistently TA- and had long, heterogeneous telomeres typical of ALT. Further study of the ALT-Low phenotype is warranted because the low levels of ALT markers may result in this phenotype being overlooked and tumors misclassified.

TMM-DP cell lines also exhibited a wide range of TMM-related parameters, including mean telomere lengths that spanned from

**Fig. 7 | Distinct gene dependencies associated with ALT and telomerase activity. A** Differential comparison of gene dependencies determined from CRISPR/Cas9 whole-genome screens of ALT (including ALT-Low) and TEL cell lines. Tissue type was used as a covariate (*N* = 521). **B** Gene dependency scores of representative preferentially essential genes (PEGs) for ALT cell lines. A higher dependency score indicates higher gene essentiality. Box plots show the 25th and 75th percentiles (box edges), median (horizontal line), and error bars defining 1.5x the interquartile range (IQR). N indicates the number of cell lines, *P*-values were determined by a two-sided Wilcoxon rank-sum test. Data is provided in Source Data. **C** Over-representation analysis of PEGs in ALT (top panel) and TEL (bottom panel) groups from (**A**) showing Gene Ontology (GO) terms for Biological Processes where FDR < 0.05. **D** Normalized two-sided Pearson's correlation analysis between TA and gene dependency scores in cell lines from all TMM groups accounting for tissue lineage (*N* = 540). Shading indicates the 95% confidence interval. **E** Over-representation analysis of PEGs correlating with TA from (**D**) showing GO Biological Processes with a significant negative (top panel) or positive correlation (bottom panel), respectively (FDR < 0.05). **F** Sub-mitochondrial localized proteins encoded by PEGs from (**D**) correlating with high TA. **G** Normalized two-sided Pearson's correlation analysis between TC and gene dependency scores in non-ALT cell lines, accounting for tissue lineage as a covariate (*N* = 525). In (**A**), (**D**) and (**G**) the dashed line indicates adjusted *P*-value = 0.05 determined by BH method using *P*-values from two-sided tests. *ALT includes ALT-Low. Detailed results are provided in Supplementary Data 6.

below the 10[th] percentile of TEL cell lines to above the 90[th] percentile of ALT cell lines. The low frequency and intensity of APBs and ALT-FISH foci in DP cell lines resembles the partial suppression of ALT phenotypic markers observed following exogenous expression of telomerase in ALT cells[43–46]. The incidence of TMM-DP cancers is uncertain since relatively few studies of cancer tissues have applied assays for both TEL and ALT. Identification of TMM-DP cancer cell lines in this study provided opportunity to investigate intratumoral TMM heterogeneity in vitro. Here, we show that TMM-DP cell lines were either a mixture of ALT and TEL single cells (TMM-Mixed) or comprised of individual subclones with both TMMs activated (TMM-Dual). The TMM-Dual cell line (SCH) provides compelling evidence that cancers may activate both TMMs. SCH is derived from a gastric choriocarcinoma[47], a rare cancer type that is not represented in the panel by any other cell lines. While there is no clear evidence that activating more than one TMM is advantageous for achieving unlimited replicative potential, multiple lines of evidence suggest that the TERT subunit of telomerase confers pro-tumorigenic functions independent of TEL-mediated telomere lengthening. These findings are consistent with the hypothesis that a cancer cell that initially becomes immortalized through ALT activation may acquire an additional survival or growth advantage through subsequent activation of TEL[48,49]. Further analysis of TMM-DP cell lines may provide insight into their biology, including the role of ALT and TEL in individual cells from TMM-Dual cell lines, and the response of TMM-DP cancers to TMM-targeted therapeutics.

TMM-DN tumors have been reported in several previous studies[4,50–59]. Although false negative assay results may have contributed to these reports, our identification of TMM-DN cancer cell lines that exhibited telomere shortening provides further functional evidence of cancer-derived cells that lack effective TMMs. These results support the notion that activation of a TMM and cellular immortalization is not always required for cancer development[6,7,51]. Some cell lines that were initially TMM-DN subsequently activated TEL, preventing further telomere erosion. For studies of telomere biology, these results underscore the need to document proliferative history and determine the TMM status of cell lines at the point they are investigated.

Our analysis of proteomic and transcriptomic data revealed that ALT and TEL cancer cell lines have distinct molecular characteristics with potential therapeutic implications. TEL cells showed upregulation of molecular processes involved in mitosis, DNA repair, and ribonucleoprotein complex biogenesis, while ALT cells featured pathways associated with advanced stages of tumor progression, metastasis, and inflammation. The pathways associated with TA levels were consistent with the empirically documented correlation between TA and cell proliferation, whereas ALT expression profiles are consistent with the increased metastasis and poorer patient outcomes generally observed in ALT cancers[60,61]. Transcriptomic and proteomic signatures of inflammation, interferon response, NFκβ, and cytokine signaling in ALT cells are consistent with EMT and strongly suggest activation of the cGAS-STING pathway. cGAS-STING signaling is activated during the innate immune response to cytosolic dsDNA generated by

chromosomal instability and viral infection. Notably, extra-chromosomal telomeric repeat (ECTR) DNA, which is abundant in ALT cells, also activates cGAS-STING[62–64]. In apparent contrast to our results, it was previously reported that the cGAS-STING pathway is suppressed in ALT cell lines[62]. In this context, we note that TOV112D was classified in the previous investigation as an ALT cell line but was found to be a TEL cell line with high TC in our study. The mechanisms underpinning the preponderance of inflammatory phenotypes and upregulation of cytokines detected in ALT versus TEL cancer cell lines in the current study warrant further investigation.

Differential analysis of the transcriptomic and proteomic data also dichotomized classes of chaperones that predominate in ALT versus TEL cell lines. TA correlated with the abundance of HSP90s and HSP90 co-chaperones with known roles in TERT folding and telomerase biogenesis, as well as the chaperonin family of proteins that comprise the TRiC complex involved in trafficking telomerase[65,66]. In contrast to the TEL cell lines, ALT cell lines had a relative abundance of chaperones from the sHSP and HSP40 families that facilitate sequestration of misfolded proteins to prevent their aggregation during the cellular stress response[67,68]. Differential upregulation of these HSPs in ALT cells suggests heightened proteo-toxic stress relative to TEL cell lines.

With the widespread use of molecular profiling in cancer diagnosis and clinical care, the ability to determine TMM status from 'omic data would provide valuable opportunities for stratifying patients for TMM-directed treatments and surveying TMM in large numbers of cancers. While the present study highlights the limitations of using *ATRX/DAXX* status, *TERT* promoter status, and TERT RNA to infer TMM, previous studies have used these features in combination with TC and telomere variant repeat (TVR) data to develop ALT classification tools[69,70]. Here, we derived an accurate ALT classifier, as well as a TA Score by applying machine learning to integrated protein, RNA, and TMM data. Our proteo-transcriptomic ALT classifier produced the strongest results in cross-validation with an AUC of 0.99, while the transcriptomic classifier and the RNA-based TA score were successfully applied to external data sets. Although external validation of the proteo-transcriptomic classifiers was limited by the lack of suitable datasets, our results demonstrate the potential diagnostic utility of 'omic tools trained on TMM data for predicting ALT cancers and TA levels. New datasets that include TMM, transcriptomic and/or proteomic data will be valuable for further testing these classifiers, including their capacity to predict ALT and TA from patient samples.

Analysis of gene dependency data from whole genome knockout screens identified sets of genes that are preferentially essential for ALT or TEL cell lines, as well as PEGs related to TA levels and TC. A subset of the ALT PEGs that function in the resolution of replication stress were previously identified as molecular targets in ALT cells, particularly *SMARCAL1* and *FANCM*[71–74]. ALT PEGs identified here that warrant further investigation as potential therapeutic targets include *SAMHD1*. *SAMHD1* encodes a dNTP triphosphohydrolase that regulates the availability of dNTPs for reverse transcription and replication, which was recently shown to impact TEL-mediated telomere lengthening[75]. However, the dependency of ALT cells on this gene may also be linked

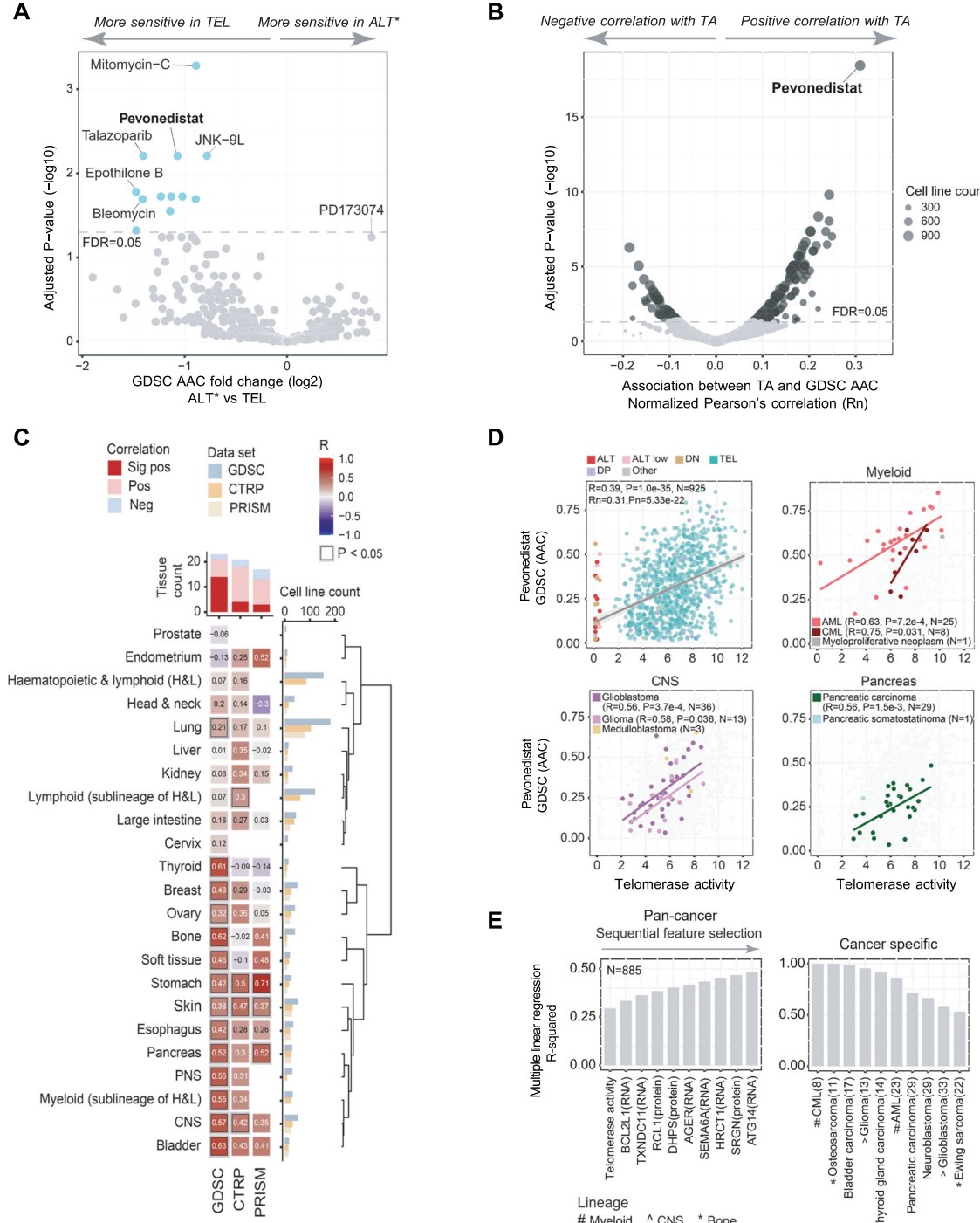

**Fig. 8 | Correlation between TA and sensitivity to Pevonedistat. A** Comparison of drug response (GDSC dataset) in ALT (including ALT-Low) cell lines versus TEL cell lines, using tissue type as a covariate. The analysis covers 636 drugs screened against 204 to 903 cell lines per drug, as detailed in Supplementary Data 7. **B** Normalized two-sided Pearson's correlation analysis of GDSC drug response and TA with tissue type regressed out. Results are shown for 936 cell lines and 644 drugs, where drug results were available for at least five cell lines. In (**A, B**), the dashed line indicates adjusted $P$-value = 0.05 determined by the BH method using $P$-values from two-sided tests; Sample number details are in Supplementary Data 7; AAC: Area above the curve. **C** Correlation values from two-sided Pearson's analysis from tissue level analysis of TA level and Pevonedistat AAC from GDSC, CTRP, and PRISM datasets using data from drugs tested on at least five cell lines in our panel. The number of cell lines included in each drug screen is detailed in Supplementary Data 7. The column graph at the top of the heatmap shows the number of tissue types tested for each drug dataset and summarizes the number of positive (Pos), negative (Neg), and

significant positive (Sig pos) correlations where $P < 0.05$. The number of cell lines in each tissue type is indicated by the horizontal bar graph at the right of the heatmap. **D** Association between TA and AAC for Pevonedistat (GDSC) at the pan-cancer level and in cancer types from the specified tissue lineages. The shaded area in the top left panel indicates the 95% confidence interval. N indicates cell line sample number. Data is provided in Source Data. **E** Correlation between Pevonedistat sensitivity (GDSC) and up to 10 features derived by SFS from a dataset of 43,366 features encompassing TMM (TA, CC, and TC), transcriptomic (RNA intensity), proteomic (protein abundance) and genomic (mutations) data. The left panel shows the coefficient of determination (R-squared) for the top feature (TA) and multiple linear regression models that sequentially incorporated features up to the top 10. The right panel shows correlations between Pevonedistat AAC and the 10 features in cancer types from tissues with robust correlations between Pevonedistat and TA. $N = 885$, tissue lineage was used as a covariate in the regression analyses.

to SAMHD1's dNTPase-independent role in DNA end resection during homologous recombination-mediated repair[30,76]. Another ALT-PEG identified herein is *ETAA1*. *ETAA1* has a known role in replication stress, activating ATR to facilitate restart and progression of the replication fork[77,78]. The ALT-PEG *SUB1* is a transcriptional co-activator that may also function at replication forks in telomeres, as it has been shown to interact with G-quadruplexes implicated in replication fork stalling[79]. *SIVA1*, *CHTF18*, and *DSCC1* are ALT-PEGs identified here that have related functions in the DNA damage response and DNA replication[80,81]. ALT cell lines also showed dependency on *TERF2IP*, a shelterin component with a key role in telomere protection mediated by suppression of homology-directed repair[82].

Among the non-ALT cell lines, high TC and high TA were associated with gene dependencies relating to mitochondrial gene expression and function, heightened cellular respiration, and metabolism, consistent with a higher proliferative rate. Conversely, TC was inversely correlated with dependence on the shelterin component TERF1 and CST complex proteins. While the association between TC and CST dependence was previously reported[12], our study further demonstrates that this association applies in both ALT and non-ALT cell lines. Our study also shows that non-ALT cell lines with low TC are more vulnerable to loss of POT1 and ACD, consistent with the role of these two shelterin components as critical regulators of telomerase processivity[83]. Collectively, the results indicate that cancer cells with long telomeres are less dependent on the CST complex irrespective of the TMM and that cells with short telomeres have heightened vulnerability to loss of telomerase processivity factors.

This study demonstrates a positive correlation between TA levels and sensitivity to the investigational drug Pevonedistat (MLN942). Pevonedistat is a first-in-class inhibitor of NAE1 that promotes neddylation of cullin-ring E3 ligases and subsequent proteasomal degradation of target proteins. NAE1's downstream targets include proteins involved in DNA replication, the DNA damage response, and various signaling pathways[36]. Although no compelling link was identified between TA and the known Pevonedistat target pathway, the observation that Pevonedistat induces DNA re-replication during S-phase, causing cell death through a G2 checkpoint response[36,84], is concordant with the sensitivity of cell lines with high TA and correspondingly high proliferation rates to Pevonedistat. Our results showed a particularly strong correlation between TA and Pevonedistat sensitivity in cell lines from the central nervous system, bladder, pancreas, and myeloid lineage leukemias. Pevonedistat is currently being tested in Phase I-III clinical trials for various solid tumors and hematologic malignancies (https://clinicaltrials.gov/), with mixed outcomes reported for patients with AML or high-risk myelodysplastic syndrome. Our results showing that Pevonedistat activity correlates more closely with TA than NAE1 and its downstream mediators suggest that a measure of TA, such as the multi-omic TA Score derived in this study, may be a valuable biomarker for guiding Pevonedistat treatment in future trials of specific cancer types.

Overall, the discoveries made in this study illustrate the extensive potential for leveraging the TMM resource we have generated to exploit telomere biology in investigations of new therapeutic and diagnostic approaches to cancer treatment. The detailed characterization of ALT cell lines in a range of cancer types and the identification of TMM states that may be mistaken for canonical ALT if characterization is less complete, are important steps toward this goal. These findings are particularly significant given the unmet therapeutic need for the broad range of cancer types that utilize ALT.

## Methods
### Cell culture
Cell lines were cultured in the media indicated in Supplementary Data 1 in a humidified incubator at 37 °C with 5% CO2. All cell lines were authenticated by 16-locus short-tandem-repeat (STR) profiling and

tested for mycoplasma contamination by CellBank Australia (Children's Medical Research Institute, Westmead, NSW, Australia). Cell lines listed in the ICLAC Register of Misidentified Cell Lines v13 [https://iclac.org/databases/cross-contaminations/] are indicated in Supplementary Data 1. STR profiling and mycoplasma tests were repeated at the end of long-term cultures. Where indicated, single cells were sorted from cell populations suspended in 2% (fetal bovine serum) FBS and 100 ng/ml DAPI (Sigma-Aldrich) in phosphate-buffered saline (PBS) using a BD Influx flow cytometer. Viable cells were selected as DAPI-negative, and single cells, gated as trigger pulse width low, were dispensed into standard culture medium in 96-well plates.

### C-Circle (CC) assay
Rolling circle amplification and detection of C-circles (CC) was performed essentially as described[85]. Cell pellets were lysed in QCP buffer (50 mM KCl, 10 mM Tris-HCl pH8.0, 2 mM MgCl₂, 0.45% NP40, 0.45% Tween-20 with 50 mAU Qiagen Protease/100 ml QCP buffer). Samples were assayed in duplicate with a negative control sample without Φ29 polymerase. Lysates (10 μl) containing the equivalent of 20 ng DNA were combined with 10 μl 0.2 mg/ml BSA, 0.1% Tween, 4 mM DTT, 1 mM each dATP, dGTP, dCTP and dTTP, 1× Φ29 Buffer and 7.5 U Φ29 DNA polymerase (NEB) and incubated at 30 °C for 8 h then 65 °C for 20 min. Reaction products were diluted to 100 μl with 2× SSC and dot-blotted onto a 2× SSC-soaked Biodyne B nylon membrane (Pall Corporation). The membrane was air-dried, crosslinked twice with 1,200 J 454 nm UV-C irradiation (Stratagene UV Stratalinker 1800) and hybridized overnight at 37 °C with end-labelled $^{32}$γP-(CCCTAA)$_3$ and PerfectHyb Plus hybridization buffer (Sigma). Following hybridization, the membrane was washed three times in 0.5x SSC/0.1% SDS at 37 °C and exposed to a storage phosphor screen. The phosphor screen was imaged using an Amersham™ Typhoon™ laser scanner platform (Amersham Typhoon control software, version 3.0.0.2, Cytiva) and densitometry analyzed using ImageQuant software (version 8.1.0.0, Molecular Dynamics). CC levels were calculated as the difference in the intensity of samples incubated with and without Φ29 polymerase, then normalized to the results from DOS16 in each assay. Five additional well-characterized ALT cell lines (U-2-OS, SAOS-2, G292-CloneA141B1, SK-N-FI and SK-LU-1) and six known TEL cell lines (HT-1080, HCT-116, A549, MCF7 and HL-60) were used as additional references for setting thresholds for positive and negative results. Values are shown as a percentage of DOS16.

### Measurement of Telomerase activity (TA)
Telomerase activity (TA) was measured in each cell line using an adapted two-step SYBR Green real-time telomeric repeat amplification protocol (qTRAP)[86]. Cell pellets were lysed in qTRAP lysis buffer (10 mM Tris−Cl pH 8.0, 1 mM MgCl₂, 1 mM EGTA pH 8.0, 10% v/v Glycerol, 0.5% w/v CHAPS detergent, 1 mM phenylmethylsulfonyl fluoride (PMSF), 15 mM β-mercaptoethanol). Following microcentrifugation, cleared cell lysates containing 5 μg total protein were added to the telomere extension buffer (1 mM dNTP mix (Thermo Fisher Scientific), 150 ng M2 primer (5'-AAT CCG TCG AGC AGA GTT −3'), 40 units RNaseOUT™ Recombinant Ribonuclease Inhibitor (10777019, Thermo Fisher Scientific), 20 mM 10 mM Tris−Cl pH 8.0, 1.5 mM MgCl₂, 63 mM KCl, 0.05% v/v Tween-20, 1 mM EGTA pH = 8.0). The reactions were incubated at 30 °C for 30 min for the telomere extension step, then heated at 95 °C for 20 min. For qPCR amplification of telomeric DNA, 1 μL telomere extension product was added to a 10 μL reaction system with 50 ng M2 primer, 25 ng ACX primer (5'- GCGCGGCTTACCCT TACCCTTACCCTAACC −3') and 1x SensiFAST™ SYBR® No-ROX Master Mix (Bioline). Each sample was run as technical triplicates using the following qPCR cycle: 95 °C for 5 min, then 45 cycles of 95 °C for 15 s, 60 °C for 60 s and 72 °C for 30 s. Negative controls in each assay included reactions with samples exchanged for water, lysate from the ALT cell line U-2-OS, and heat-inactivated controls for every sample

(95 °C for 20 min). Average values were calculated relative to a standard curve generated using serial dilutions of lysate from the MCF7 cell line. TEL cell lines HeLa, HT-1080, HCT-116, A549, MCF7 and HL-60 were used to calibrate the assay. TA values are presented as a percentage of MCF7.

The Direct TA assay was used to verify telomerase levels in a subset of cell lines according to the published method[87]. Briefly, snap frozen pellets of $10 \times 10^6$ cells were resuspended in 1 mL cold Buffer A (50 mM HEPES-KOH.pH8.0; 300 mM KCl; 2 mM MgCl$_2$; 10% v/v glycerol; 0.1% v/v Triton X-100, supplemented with PMSF, and rotated at 4 °C for 1 h. Lysates were clarified by centrifugation, treated with 20 µg/ml anti-hTERT antibody for 30 min, and incubated with 25 µL Protein G/sepharose beads (25 µL, 1 h). Beads were washed with Buffer A, resuspended in Buffer A with 1 mM DTT, and eluted with peptide antigen (10 µL, 30 min, RT). Immunopurified telomerase (80 µL) was combined with extension master mix (final concentrations 300 mM KCl, 10 mM DTT, 2 mM MgCl$_2$, 1 µM 5'-(CTAGACCTGTCATCA)$_3$-Biotin-3', 1 mM dATP, 1 mM dTTP, 10 µM dGTP, 40 µCi α-$^{32}$P-dGTP) and incubated at 37 °C overnight. The reaction was quenched with Bind/Wash buffer, captured on Dynabeads, washed, and denatured at 80 °C. Products were resolved on high-resolution polyacrylamide gel and quantified by phosphor imaging. TA measured by the Direct Assay is presented as a percentage of the A549 cell line. All samples were assayed as biological replicates from two time points.

### Telomere content (TC) qPCR
Telomere content (TC) qPCR was performed on a Rotor-Gene Q real-time cycler (Qiagen) under monochrome multiplex conditions adapted from Dahlgren et al. 2018[88]. Each 25 µl reaction contained 1x Rotor-Gene SYBR Green (Qiagen), 5 ng of DNA template, and primer sets. The final concentrations for the primers were: (i) 300 nM each of telomere forward (CGGCGGCGGGCGGCGCGGGCTGGGCGGCGGTTTGTTTGG GTTTGGGTTTGGGTTTGGGTTTGGGTT) and reverse (GCCCGGCCCG CCGCGCCCGTCCCGCCGGGCTTGCCTTACCCTTACCCTTACCCTTAC CCTTACCCT) and (ii) 400 nM each of the Multiple Copy Sequence (MCS) forward (GGTGATGGGATTTCCATTGATG) and reverse (CTTCATTGACCTCAACTACATGG) primers. The MCS primers amplify 116 to 119 bp products from at least 6 loci on chromosomes 1, 5, 8, 12, and 19. The qPCR cycle was as follows: Stage 1: 5 minutes at 95°; Stage 2: 2 cycles of 15 s at 94°, 15 s at 49°; Stage 3: 35 cycles of 15 s at 81°, 30 s at 60° with signal acquisition for MCS; Stage 4: 5 min for 95°; Stage 5: 35 cycles of 15 seconds at 95°, 30 s at 84° with signal acquisition for telomere. Data was analyzed using Rotor-Gene Q software 1.7 (Qiagen). Telomere products were normalized to the MCS products and results expressed as the average of triplicate reactions. The telomere content was calculated relative to the TEL cell line HT-1080 set at an arbitrary value of 8.0.

### Terminal restriction fragment (TRF) Southern blot analysis
Genomic DNA was extracted from cell pellets using QIAamp DNA Blood Mini Kit (Qiagen) and digested with restriction enzymes HinfI (2.5 U/µg DNA) and RsaI (1.5 U/µg DNA) in 1x CutSmart buffer (New England Biolabs). Digested DNA was resolved on a 1% agarose gel using a CHEF-DR II pulsed-field electrophoresis apparatus (Bio-Rad). The gel was dried, denatured, and hybridized to $^{32}$P-ATP-labeled (TTAGGG)$_4$ telomeric probe in Church buffer containing 0.5 M Na$_2$HPO4 (pH 7.2), 1 mM EDTA, 7% SDS and 1% BSA. The gel was washed three times in 4x SSC at room temperature and once at 50 °C before exposing to a storage phosphor screen for approximately 7 days. The phosphor screen was imaged using an Amersham™ Typhoon™ laser scanner platform (Cytiva). For calculation of mean terminal restriction fragment (TRF) length, ImageQuant software (Molecular Dynamics) was used to quantify signal intensity across a grid of 150 segments per lane. Background was subtracted from the signal intensity of each segment and mean telomere length calculated as $(\Sigma(S_i \times L_i))/(\Sigma S_i)$ where $S_i$ is TRF

signal at a given location after background subtraction, and $L_i$ is the corresponding DNA fragment length determined relative to molecular weight markers[89]. Uncropped gel images are shown in Source Data and Supplementary Information.

### Western blot analysis
Cell pellets were lysed in RIPA buffer (Cat#89901 PierceTM Thermo Scientific) containing 1x Complete™ Protease Inhibitor Cocktail (Cat#04693132001 Roche) at a concentration of $1 \times 10^6$ cells/60 µl. Protein was quantitated with the PierceTM BCA Protein Assay Kit (Cat#23225 Thermo Scientific) and 30 µg of protein per sample was resolved by electrophoresis through NuPAGE 3–8% Tris-Acetate Gels (Cat#EA0378BOX Invitrogen) for ATRX and DAXX detection and NuPAGE 4–12% Bis-Tris Gels (Cat#NP0336BOX, Invitrogen) for SAMHD1 detection. Following electrophoresis, proteins were transferred to Immobilon-P PVDF Membrane (Cat#05317 Millipore). Membranes were blocked overnight in 5% skim milk in 1x Phosphate-Buffered Saline, 0.1% Tween20 (PBST), incubated for 3 h at room temperature (RT) with primary antibody, washed three times in PBST, incubated with secondary horseradish peroxidase (HRP)-conjugated antibody for 1 h and washed three times in PBST. Proteins were detected with SuperSignalTM West Pico PLUS chemiluminescent substrate (Cat#34578 Thermo Scientific) and visualized using the GE ImageQuantTM LAS-4000 luminescent imager analyzer (LAS-4000, Fujifilm). Primary antibodies: anti-ATRX at 1:400 dilution (Cat#HPA064684 Sigma), anti-DAXX at 1:750 dilution (Cat#HPA008736 Sigma), anti-SAMHD1 at 1:500 dilution (Cat# TA502024, OriGene Technologies Inc.) and anti-Actin at 1:500 dilution (Cat#A2066 Sigma). Secondary antibodies: Goat Anti-Rabbit Immunoglobulins/HRP (1:5000) (Cat# P0448 Agilent) and Goat Anti-Mouse Immunoglobulins/HRP (1:5000) (Cat# P0447 Agilent). HeLa and U-2-OS cell lines were used as controls for ATRX and DAXX Western blots. Uncropped images of membranes are shown in Source Data or Supplementary Information.

### Immunofluorescence assays for alternative lengthening of telomeres
Whereas the CC assay detects partially double-stranded circles of telomeric DNA in cells that have activated the Alternative Lengthening of Telomeres (ALT) mechanism[90], the ALT-fluorescence-in-situ-hybridization (FISH) assay detects single-stranded C-rich telomeric DNA[91] and the ALT-associated PML Body (APB) assay provides an independent indicator of ALT phenotype by detection of co-localized PML and telomeric DNA[92]. Cells for immunofluorescence assays were resuspended in PBS and cytocentrifuged onto SuperFrost Plus Adhesion Microscope Slides (Menzel-Glaser) at 72 x g for 10 min in a Shandon Cytospin 4 at medium acceleration. APBs were detected as described[92]. Briefly, slides were fixed in 4% (v/v) formaldehyde in PBS, permeabilized with KCM buffer (120 mM KCl, 20 mM NaCl, 10 mM Tris pH 7.5, 0.1% Triton X-100), washed in PBS, and blocked in 100 µg/mL DNase-free RNase A (Sigma) in antibody dilution buffer (20 mM Tris pH 7.5, 2% BSA, 0.2% Fish Gelatin, 150 mM NaCl, 0.1% Triton, 0.1% Sodium Azide). Slides were incubated with goat anti-PML (N-19) antibody (Santa Cruz, catalogue # sc-9862) at 1:500 dilution and then incubated with Alexa Fluor 488 donkey anti-goat secondary antibody (Invitrogen, catalogue # A-11005) at 1:500 dilution. Slides were then cross-linked with 4% formaldehyde and dehydrated with ethanol. Telomeric DNA was detected using the TelG-TAMRA (TTAGGG)$_3$ PNA probe (Panagene, catalogue # F2002).

The ALT-FISH assay was performed on cells that were cytocentrifuged onto slides as described[91]. After pre-extraction in permeabilization buffer (20 mM Tris-HCl pH 8.0, 50 mM NaCl, 3 mM MgCl$_2$, 300 mM sucrose, 0.5% Triton X-100) and fixation in 4% (v/v) formaldehyde in PBS and pre-extracted again, the slides were treated with RNAse A (100 ug/mL) for 1 h at room temperature, ethanol dehydrated

and incubated with TelG-TAMRA LNA probe (Qiagen, GeneGlobe ID: YCO0284647 Catalog # 339411) for 1 h at room temperature. Slides were then washed and dehydrated in ethanol.

Slides from APB and ALT-FISH assays were imaged and captured as Z-stacked images on Zeiss Axio Imager Z2 (Zeiss). At least 500 nuclei per cell line were imaged in DAPI, TAMRA and 488 channels at fixed exposure time and light intensity settings at 63x. Images were processed into extended depth of focus projections using ZEN software (Zeiss), then imported into CellProfiler v4.2.5[93] for automated image analysis. The DAPI channel was used to identify individual nuclei, then foci within each nucleus were masked using an intensity-based threshold. An APB was defined as a colocalization where at least 75% telomere DNA overlapped PML protein signal. The number of nuclear foci in the ALT-FISH assay and co-localized signals in the APB assay, as well as signal intensities, were quantified. APB and ALT-FISH Indexes were calculated as the product of the frequency of cells with >2 ALT-FISH foci or APBs per nucleus and signal intensity. Cut-offs for positive and negative results were set relative to characterized ALT and TEL cell lines.

## Proteomics

The proteomic data were acquired by data-independent mass spectrometry (DIA-MS) from 2-3 replicate samples. Cells were harvested from semi-confluent cultures of T75 flasks and lysates prepared using our previously published accelerated barocycler lysis and extraction method with clean-up on solid phase extraction Waters HLB cartridges[94]. Cell lysates (11 μl) were mixed with an equal volume of 6% (w/v) sodium deoxycholate and then subjected to additional HnB processing steps, clean up on Waters MCX cartridges, and Liquid chromatography-mass spectrometry in DIA mode as described[8]. DIA data was searched with DIA-NN (1.8.1)[95] except that no missed cleavages were allowed when constructing the library. The False Discovery Rate (FDR) was set at 1% and only proteotypic peptides were used for quantitation. The DIA-NN normalized peptide matrix was subsequently rolled up to the protein level using maxLFQ via the 'diann_maxlfq' function in the DIA-NN R package[96]. Protein intensities were log$_2$-transformed, and average values from replicates were applied in downstream analyses. To circumvent the missing value problem, proteins were retained that were quantified in more than 10% of all samples, or more than 30% of samples in either ALT or DN groups, and missing data were imputed with the minimal intensity of individual proteins for further analyses. Non-imputed proteomic data were employed in Pearson's correlation analysis to preserve true variability.

## Whole genome sequencing

Whole genome sequencing of 14 ALT and ALT-Low cell lines was performed to obtain 150 bp paired-end reads on an Illumina NovaSeq Instrument at the Australian Genome Research Facility, Melbourne, VIC. Raw sequencing reads were preprocessed according to the GATK4 best practices (GATK version 4.4.0.0)[97–99] using the nf-core/sarek (v3.4.0) analytical pipeline[100,101] and Nextflow (v23.10.1)[102]. Adapter trimming and quality control were performed using FastP (v0.23.4)[103] and FastQC (v0.12.1), respectively. Trimmed reads were aligned to human GRCh38 reference genome using bwa-mem (v0.7.17-r1188)[37]. Aligned reads were processed using GATK4 MarkDuplicates, followed by base quality score recalibration using BaseRecalibrator and ApplyBQSR.

Structural variants (SVs) were called using three algorithms, namely Manta[104] in tumor-only mode, Smoove, and TIDDIT[105], using default parameters. SURVIVOR[106] was used to merge the single-caller VCF files and retain SVs supported by at least two SV callers. The SV events of longer than 40 bp, with a maximum distance of 1 kb, and same strand and type were merged and annotated using the AnnotSV[107]. Samplot[108] was used for visualization and manual curation of structural variants. Analysis of copy number variation (CNV) was performed using the

tumor-only mode of cnvkit (v0.9.10)[109] implemented in nf-core sarek (v3.4.0) with default settings. The segment's genomic coordinates and log$_2$ ratio values were used to estimate gene-level copy numbers. Single-nucleotide variations (SNV) and short insertions and deletions (indels) were called using the tumor-only mode of nf-core sarek Mutect2 (GATK v4.4.0.0)[110] with default settings and filtered using FilterMutectCalls. In Mutect2 analysis, the GATK panel-of-normals and gnomAD allele frequencies were used as the PoN and germline resource, respectively.

## Multi-omic data analysis

Genomic data, including exonic SNVs, exonic indels, and CNVs, were extracted from whole-exome sequencing and Affymetrix SNP6 arrays available through Cell Models Passport[34]. Gene fusions predicted from RNA-Seq data were retrieved from Cell Models Passport. Values for CNV were rescaled using −1.5 to 1.5 as limits. SV data with SvABA calls[9] were downloaded from the DepMap portal[39]. Results for mutations, SVs, and fusions in *ATRX* and *DAXX* genes were plotted if at least one variant was detected in the cell line subpanel. *TERT* and *TERC* gene expression data were derived from RNAseq data made available through Cell Models Passport. Raw RNA read counts downloaded from the "rnaseq_read_count_20220624" file were reprocessed into normalized log$_2$-counts per million (logCPM) for downstream analysis using a previously described approach[111]. Processed data from bisulfite conversion sequencing for methylation of the *TERT* hypermethylated oncological region (THOR) were retrieved from a previous publication[112]. THOR methylation values used in this study are average values of 24 CpG islands within the previously defined 433 bp region (GRCh37:5:1295321-1295754, GRCh38:5:1295206-1295639)[18]. Stemness Index and EXTEND scores were derived using logCPM RNA expression data according to previously published methods[23,26].

## Gene dependency analysis

Gene dependency analysis was performed using the integrated CRISPR-Cas9 screen dataset made available through the Project Score portal[29]. This dataset merges the genome-wide essentiality measurements from the Wellcome Sanger Institute's (WSI's) Project Score and Broad Institute's Project Achilles, covering 17,486 genes and 540 cell lines from our panel, and quantifies gene dependency as the CERES Index[113]. Analysis of gene dependencies in the present study applied minus CERES Index (- CERES Index) such that a higher score indicates a higher likelihood that the gene is essential in a given cell line, a 0 score indicates a gene is not essential and a score of 1 is the median of all pan-essential genes.

## SAMHD1 gene suppression analysis

Eight cancer cell lines were subject to siRNA-mediated suppression of *SAMHD1*; Hs-746T (gastric carcinoma), CAL-72 (osteosarcoma), SAOS-2 (osteosarcoma), HuO9 (osteosarcoma), HGC27 (gastric carcinoma), SJSA-1 (osteosarcoma), 143B (osteosarcoma) and HT-1080 (fibrosarcoma). The cells were seeded at 10% to 20% confluency in triplicate wells in 96-well plates for proliferation and cell viability assays, and in six-well plates for Western blot analysis of knockdown efficiency. The following day, the cells were transfected with 25 nM siRNA in media containing Lipofectamine™ RNAiMAX (Thermo Fisher Scientific) at 0.2% (v/v), following the manufacturer's protocol. The siRNAs employed were siSAMHD1(I) (s24792, Thermo Fisher Scientific); sense sequence: CGCAACUCUUUACACCGUAtt; anti-sense sequence: UACGGUGUAAA GAGUUGCGag, siSAMHD1(II) (s533348, Thermo Fisher Scientific); sense sequence: GUGCUAAACCCAAAGUAUUtt; anti-sense sequence: AAUA CUUUGGGUUUAGCACtg, and a scrambled siRNA control (#4390847, Thermo Fisher Scientific). The culture medium was refreshed within two days after transfection. A growth curve was generated using confluency measures captured at 4-hour intervals over 7 days using an Incucyte® ZOOM system (version 2018A). For quantification of proliferation, the area under the curve (AUC) was quantified using GraphPad Prism

(version 10.6.1). The CellTiter-Glo® Luminescent Cell Viability Assay (Promega) was used following the manufacturer's instructions to measure the viability of cells in 96-well plates 72 h after transfection. Cells were harvested from the 6-well plates for Western blot analysis 72 h after transfection.

### Drug sensitivity data analysis

Drug sensitivity analysis employed the GDSC1 and GDSC2 datasets produced by WSI, covering 644 drugs/drug combinations tested on 936 of the 976 cell lines in the present study panel[13,33,34,114]. For the compounds that were screened in both GDSC1 and GDSC2, we used only the GDSC2 data, as recommended at the GDSC portal ( https://www.cancerrxgene.org/). Area above the curve (AAC), which equals 1-AUC (area under the drug concentration-time curve) for the GDSC datasets, was used as the measure of drug response, where a higher AAC indicates greater drug sensitivity. AAC measurements from the independent drug sensitivity screening project CTRP[37,38] and PRISM[115], performed by the Broad Institute, were downloaded from PharmacoDB database[116].

### Differential and correlation analysis

Differential comparisons of gene and protein expression, gene dependencies, and drug sensitivity in different TMM groups were performed using the base package Stats in R (version 4.2.3)[117], applying a two-sided linear regression model with tissue type as a covariate. Pearson's correlation evaluation and linear regression were also performed using the Stats in R package. Normalized correlation tests applied two-sided Pearson's method utilizing the residuals of both target features after linear regression on tissue type to adjust for potential tissue effects. Log-transformed TMM measurements ($Log_2(n+1)$), including TA, CC, and TC, were utilized for analyses. Linear regression was also conducted via Stats package. For differential comparisons and linear regression, the adjusted P-value was determined by the Benjamini-Hochberg (BH) method for controlling the false discovery rate (FDR) from two-sided testing.

### Gene set enrichment (GSEA) and over-representation analysis (ORA)

Gene set enrichment analysis (GSEA) and over-representation analysis (ORA) were performed using WebGestalt 2019 tool [webgestalt.org][118]. Input gene and protein lists for GSEA were ranked according to $log_2$ expression fold change or the correlation coefficient value. For input lists of greater than 100 genes, redundancy reduction processing was performed using the Weighted Set Cover filtering option, and the "noRedundant" version of Gene Ontology (GO) annotation was selected as the reference. GO biological process and chaperone categories, and Hallmark (Broad Institute) data sets were used as references. Enrichment analysis focusing on human molecular chaperones was conducted using the gene list retrieved from ChaperoneNet database[27] as an annotation reference.

### Machine learning for the derivation of TMM classifiers and predictive TA score

ALT and TA prediction algorithms were trained on the WSI-CMRI datasets. To generate ALT-specific classifiers, the training sample set was down-sampled by excluding cancer types with no ALT or ALT-Low cell lines. DP cell lines were excluded, and DN and TEL samples were combined as a non-ALT group for binary analysis. For the generation of predictive scores for TA levels (TA Score), all available cell lines were included, regardless of TMM classification. Log-transformed TA values ($Log_2(n+1)$) were used in these analyses. Due to gaps in the availability of RNA and protein data for certain cell lines, varying numbers of cell lines were included in the RNA, protein, and combined RNA & protein models (Fig. S5A). Missing values were imputed in proteomic data to maximize protein coverage.

Sequential forward selection (SFS) was applied to transcriptomic and/or proteomic intensities using Python package mlxtend (v0.20.0) to identify the best five features for the TMM prediction models and TA Score. The multinomial logistic regression method of scikit-learn (v1.1.2) package was used for the ALT prediction models[119] using the class-weight option to mitigate bias toward the majority category (non-ALT), where: "LogisticRegression(class_weight = 'balanced', random_state=seed". Area under the receiver operating characteristics (AUROC) curve was used as the evaluation metric. For the TA Score, the best five features were used to train the TA prediction models utilizing the multiple linear regression methodology of scikit-learn (v1.1.2). Pearson's correlation coefficient derived from regression analysis of the predicted and observed TA were used as evaluation metrics. Feature importance was calculated using the Python package SHapley Additive exPlanations (SHAP) (v0.41.0)[120].

The performance of the TMM/TA prediction algorithms was evaluated by 5-fold cross-validation against the training set. The RNA-based TMM predictor was further tested against external cohorts of human cell lines, mesenchymal stem cells, and liposarcoma tissue samples with published RNA microarray and TMM data[21,22]. The RNA-based TA predictor was tested on published cohorts characterized by TA assays and transcriptomics (lung cancer cell lines[23], glioma sphere-forming cells[25], and bladder cancer cell lines[24]), and compared to TERT RNA intensity alone and to the previously published 13-gene signature EXTEND Score[23].

The association between GDSC drug sensitivity data for Pevone-distat and features selected using SFS was evaluated using multiple linear regression models with tissue type as a covariate. Drug sensitivity was analyzed as AAC, and the 10 features were selected from log-transformed ($Log_2(n+1)$) TMM measurements (TA, CC, and TC) and multi-omic data (SNVs and Indels from genomic data, RNA intensities, and proteomic abundance) from 885 cell lines that had all data types available.

### Statistics and reproducibility

The cell line panel used in this study was based on previous large-scale multi-omic and functional studies conducted by the WSI. The WSI panel was supplemented with 25 tumor-derived cell lines from Children's Medical Research Institute (CMRI) to improve representation of less common TMM states, including ALT. No statistical methods were used to pre-determine sample size. Samples were processed in randomized batches for proteomic analysis, while wet-lab analyses were performed in batches based on sample arrival time. Mass spectrometry and orthogonal validation of TA were conducted by blinded investigators. All other experiments were conducted unblinded due to the lack of prior data on the majority of the cell lines that could introduce bias.

All telomere biology metrics were obtained from technical replicates. To verify reproducibility, TA, CC, and TC data were also collected from biological replicates of subpanels of cell lines using additional time points and independently grown cell stocks at CMRI and WSI (Supplementary Fig. S1C, D). Results were also validated through correlation in orthogonal assays. Well-characterized ALT and TEL cell lines were used as controls in all telomere biology assays, Western blots, and TRF analyses. Reproducibility of Western blot and TRF results was confirmed through repeated assays. No data was excluded except for cell lines found to be misidentified or contaminated by STR profiling. Cell lines were classified as "Other" when data did not clearly align with a distinct TMM phenotype.

P-values in box plots were calculated using two-sided Wilcoxon rank-sum tests. Correlation analyses used two-sided Pearson's tests to determine R and P-values. Adjusted P-values were determined by the Bonferroni-Hochberg (BH) method for false discovery correction using P-values from two-sided testing. Statistical analyses were performed using the ggpubr (Wilcoxon rank-sum tests) and stats

(correlation analysis) packages in R (version 4.2.3), or GraphPad Prism (version 10.6.1). ROC plots and AUROC values were generated using the ROCR package in R (version 4.2.3).

## Ethics and Inclusion
The cell line panel used in this study comprised 976 cell lines, with 951 cell lines grown at the WSI, Cambridge UK, and an overlapping set of 154 grown at CMRI, Sydney, Australia (Supplementary Data 1). The cell lines were originally derived from male and female children and adults of diverse ethnicities. This study was conducted in accordance with institutional governance and ethical oversight policies.

## Reporting summary
Further information on research design is available in the Nature Portfolio Reporting Summary linked to this article.

## Data availability
Measurements from TMM assays, including qTRAP, C-Circle assay, TC qPCR, ALT-FISH, APB assays, Telomere-FISH, TC by qPCR, and Western blot analysis of ATRX/DAXX are provided in Supplementary Data 2. Raw and processed proteomic data generated in this study and accompanying files have been deposited in the ProteomeXchange Consortium via the PRIDE partner repository with the dataset identifier PXD058664. Whole genome sequencing data generated from 14 ALT and ALT-Low cell lines were deposited in the Sequence Read Archive (SRA) database under accession code PRJNA1251703. Summarized data on *ATRX* and *DAXX* mutations, SVs and CNVs are included in Supplementary Information (Supplementary Data 3).

Genomic and transcriptomic data for other cell lines were retrieved from external sources: SNV and indels (version 20221018), gene fusion (version 20191101), CNV (version 20221213), and RNA expression (version 20220624) were downloaded from Cell Models Passport [https://cellmodelpassports.sanger.ac.uk/downloads][10]. SV data (translocation SvABA version CCLE 2019) was also obtained from DepMap [https://depmap.org/portal/][9]. Telomere content data, previously derived from whole exome sequencing and whole genome sequencing, were downloaded from DepMap [https://depmap.org/portal/data_page/?tab=allData][12]. *TERT* promoter mutations and THOR methylation were retrieved from previous publications[9,112]. Gene dependency data [version: Integrated CERES_ComBat+QN+PC1] was downloaded from Figshare [https://figshare.com/articles/dataset/Integrated_Merged_Datasets/13252640?file=25521821][29]. The GDSC drug screen datasets were previously published [https://figshare.com/articles/dataset/Pan-cancer_proteomic_map_of_949_human_cell_lines/19345397?file=34355645][13]. CTRP (version CTRPv2_2015) and PRISM (version PRISM_2020) drug screen results were downloaded from PharmacoDB [https://pharmacodb.ca][116] using the PharmacoGx package in R (version 4.4.1). Data used for validation of TMM/TA predictors were from the following sources: GEO Series accession number GSE14533[21,22] for RNA expression in human cell lines, hMSCs and the liposarcoma tissue cohort; DepMap [https://depmap.org/portal/] (version 22Q4) for RNA expression in the CCLE lung and bladder cell lines[23,24]; and a previous publication for RNA expression data from Glioma sphere-forming cells and EXTEND Score data[23]. The remaining data are available within the Article, Supplementary Information or Source Data file. Source data are provided with this paper.

## Code availability
Analysis codes are available through Zenodo [https://doi.org/10.5281/zenodo.16892621][121].

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

## Acknowledgements

This work was supported by funding from Cancer Council NSW (RG24-05), the Hill Foundation, Pinnacle Investments Foundation, Sally Morrison, and the Perpetual Trustees awarded to KLM and RRR; the Australian Cancer Research Foundation, Cancer Institute NSW (2017/TPG001, REG171150), NSW Ministry of Health (CMP-01), University of Sydney, Cancer Council NSW (IG 18-01), Ian Potter Foundation, the Medical Research Futures Fund (MRFF-PD), the National Breast Cancer Foundation (IIRS-18-164) and the NHMRC of Australia's European Union grant (GNT1170739, H2020-SC1-DTH-2018-1, "iPC - individualizedPaediatricCure" (ref. 826121) awarded to RRR and PJR; the Wellcome Trust Grant (206194) to MJG; the Ernest & Piroska Major Foundation and the Kids Cancer Alliance (Australia) to SBC. We thank Ms Michele Daly for consumer input and project support. For the purpose of Open Access, the MJG has applied a CC BY public copyright license to any Author Accepted Manuscript version arising from this submission. The authors acknowledge services and technical support from the CMRI Imaging, CellBank Australia, Bioinformatics Facilities, and the Westmead Research Hub Flow Cytometry Facility.

## Author contributions

Y.W.: Method development, experimentation, data analysis, writing – original draft, manuscript review; Z.C.: Method development, data analysis, writing – original draft, manuscript review; D.C.: Method development, experimentation, data analysis, writing – original draft, manuscript review; J.R.N.: Experimentation, manuscript review; K.P.: Experimentation, manuscript review; J.L.: Experimentation, manuscript review; S.B.C.: Experimentation, manuscript review; B.E.: Experimentation, manuscript review; J.M.S.K.: Experimentation, manuscript review; RX: Experimentation, manuscript review; Z.N.: Data analysis, manuscript review; M.B.: Data analysis, manuscript review; S.V.: Experimentation, manuscript review; L.R.: Experimentation, manuscript review; SB: Supervision and project management, method development, experimentation, manuscript review; N.A.: Supervision, data analysis, manuscript review P.J.R: Funding acquisition, supervision and project management, method development, manuscript review; P.G.H.: Supervision and project management, method development, manuscript review; M.J.G.: Funding acquisition, supervision and project management, method development, manuscript review; Q.Z.: Supervision and project management, method development, manuscript review; R.R.R.: Funding acquisition, conceptualization, supervision and project management, method development, writing – original draft, manuscript review; K.L.M.: Funding acquisition, conceptualization, supervision and project management, method development, experimentation, data analysis, writing – original draft, manuscript review.

## Competing interests

R.R.R.: Consulting for Tessellate Bio BV Ltd; co-inventor on a patent for the C-circle assay. The remaining authors declare no competing interests.

## Additional information

[1]Cancer Research Unit, Children's Medical Research Institute, Faculty of Medicine and Health, the University of Sydney, Westmead, NSW, Australia. [2]ProCan®, Children's Medical Research Institute, Faculty of Medicine and Health, the University of Sydney, Westmead, NSW, Australia. [3]Cell Biology Unit, Children's Medical Research Institute, Faculty of Medicine and Health, the University of Sydney, Westmead, NSW, Australia. [4]Bioinformatics Unit, Children's Medical Research Institute, Faculty of Medicine and Health, the University of Sydney, Westmead, NSW, Australia. [5]Wellcome Sanger Institute, Wellcome Genome Campus, Cambridge, UK. ✉e-mail: qzhong@cmri.org.au; rreddel@cmri.org.au; kmackenzie@cmri.org.au

