## [Transparent Peer Review file · Nature Communications]

Large-scale drug sensitivity, gene dependency, and proteogenomic analyses of telomere maintenance mechanisms in cancer cells

Corresponding Author: Professor Karen MacKenzie

Version 0:

Reviewer comments:

Reviewer #1

(Remarks to the Author)

The manuscript "Large-scale drug sensitivity, gene dependency, and proteogenomic analyses of the spectrum of telomere maintenance mechanisms in cancer cells" conducts a large scale study of TMM in cancer cell lines. It provides a detailed atlas of proteomic and transcriptomic profiles associated with telomerase activation, and a wealth of information on TMM related assays including telomere length, TRF, TRAP, APB, and C-circle assays. The technical aspects alone will provide the field with much needed information on how the different assays compare to each other and which combination provide the potential to be a gold standard for TMM classification.

The study will provide a solid platform for the development of novel diagnostic and prognostic approaches in molecular oncology. Beyond that, important insights into TMM double positive cell lines and atypical low-ALT features cell lines are characterised, which helps to solve open questions on the interchangeability of TMMs, and thus, potential resistance mechanisms against TMM-targeting therapy.

Finally, novel insights into the sensitivity for molecular intervention are provided on the level of gene essentiality, as well as, drug sensitivity. Overall this study provides a significant advancement of the field.

Nonetheless, I remain with following comments:

Major

1. A high level comparison on how the defined TMM classes are distributed by cancer-of-origin for cancer-derived cell lines would be of interest. For instance, are ALT-low cases concentrated in particular cancer types? How do the ratio of these classes differ between, for instance, adenocarcinomas and sarcomas?
2. While the TERT promoter mutations are mostly referenced by their coordinates according to the GRCh37 genome assembly in the literature, this study uses the more recent GRCh38 coordinates. It may be helpful to provide a mapping between these two nomenclatures as courtesy to the reader.
3. The training datasets for the machine learning (ML) models have a strong class imbalance, with a more than 10-fold overrepresentation of the non-ALT, respectively, TEL cases. How does the applied ML model account for that? Was downsampling of the major class considered during the training?
4. A closer look at the poor performance of 3-way model on the Cell line & hMSC RNA test dataset may be instructive to improve the performance in future. Not surprisingly, the - in training overrepresented - TEL class performs best (nearly flawlessly). But how do the other classes fail in detail. Are rather DN and ALT confused with each other, or are they misclassified as TEL? By providing a 3x3 confusion matrix that provides details on which and how many cases are classified as which other class that can be easily clarified.
5. Among the 5 most relevant RNA features USP1 is showing up in the combined as well as the RNA model. It may be of relevance to mention that USP1 is implied in Fanconi Anemia, which creates a direct link to ALT. It would also be of interest

how it univariately relates to the TMM classes on protein level.

Minor comment:

. Figure S8 is missing the (A) label in the caption

The manuscript should be published in any case, but may be refined by addressing these comments.

(Remarks on code availability)

The code uses well established tools and should be easily adaptable to any modern computational environment. The comments provided in the code are minimal, therefore it may require an average skill level in both of the applied programming languages (R and python) to understand it.

Minor comment: Some of the code still contains hard coded pathes to the working environment of the original programmer. Optimally, these would be substituted by neutral placeholder, maybe with a comment how to setup the project on a new computer. This would include also include which directory structure is expected by the program.

Reviewer #2

(Remarks to the Author)

Using a highly powered multiomics approach, the study presents several new aspects of telomere maintenance, mostly about ALT, including putative new gene targets (SIVA1, CHTF18, and DSCC1, TERF2IP), pathways (NEDDylation, TGF) and drug pathways (NEDD inhibitors). The study reveals further aspects of ALT, including evidence that cGAS-STING are active in ALT cells and that these cells exhibit potentially constitutive inflammatory pathway signaling. By applying phenotypic analysis with the multiomic profiling of tumor cell lines, several new ALT cell lines are identified, and others are re-categorized. The authors provide salient evidence of intra-tumoral and cell-type heterogeneity of TMMs, with telomerase and ALT co-existing as double positive cellular populations. The authors also extend prior efforts (Nonneville et al.) to develop TMM classifiers, once again, focusing primarily on ALT genomic and phenotypic markers (ATRX, DAXX, CCs, FISH). Here, machine learning of proteomic/RNA data sets are used to derived TMM predictors. Overall, this work will be a useful telomere cancer genomics resource. That being said, there is a major limitation in the absence of functional studies. These would have been very important to substantiate the work and provide new advances. Thus, the studies conclusions feel somewhat underwhelming despite the very positive outcomes mentioned above. This is the primary shortcoming identified by this reviewer.

(Remarks on code availability)

Reviewer #3

(Remarks to the Author)

This MS represents an impressive amount of work. The authors provide a comprehensive resource for the telomere biology community by analyzing over 900 cancer cell lines. They classify cell lines based on their telomere maintenance mechanism using telomerase activity, C-circle assay, and telomere content. For a subset of cell lines, they also performed telomere restriction fragment analysis and cytological evaluation of ALT features. Finally, they performed extensive proteomic analysis and determined gene dependency and drug sensitivity based on available resources. The work revealed that telomere maintenance mechanisms (TMMs) in cancer are more complex than the classical view of telomerase vs. ALT. They identified double-positive (ALT+telomerase), double-negative, and mixed TMM states. They developed machine learning-based classifiers to predict TMM status and telomerase activity from 'omic data, uncovered molecular vulnerabilities (particularly in ALT cells), and associated drug sensitivities, notably identifying ALT sensitivity to FGFR inhibition and telomerase activity correlation with Pevonedistat sensitivity. Their findings provide a new resource and suggest opportunities for TMM-targeted therapies and diagnostics.

The major strength of this study is the comprehensive approach that generated what will be an invaluable resource for the telomere biology community, with a detailed analysis of 976 cell lines, integrating telomere biology with multi-omic, CRISPR, and drug sensitivity data. Furthermore, the MS provides novel insight into gene dependencies of cell lines based on the TMM and challenges the notion of a binary choice in cancer cells between telomerase and ALT positive cells. However, there are a few weaknesses that should be addressed and/or discussed.

Potential bias in cell line selection: Certain cancer types are overrepresented or underrepresented, affecting generalizability. This is particularly important for gene essentiality and drug sensitivity. While this is an intrinsic problem with this type of approach, it should be discussed more clearly in the manuscript.

Potential for confounders: Tissue-specific effects may complicate interpretation of vulnerabilities or drug responses despite attempts to control for them.

Machine learning classifiers: While this is very interesting and has clear potential therapeutic value, it is not discussed nor verified sufficiently. Classifiers depend heavily on training datasets; applicability across broader sample types (e.g., patient biopsies) needs further testing. Additionally, there appears to be a potential contradiction between Figure 5A and Figures

6A-B, as the features that appear to be the best predictors in Figure 5A do not seem selectively enriched in the biological pathways shown in 6A-B. The authors should clarify this point.

Functional validation missing: Predictions (especially from gene dependency screens) need experimental validation to confirm therapeutic relevance. While I appreciate that the authors already provided an enormous amount of data, the lack of any type of validation limits confidence in some of the suggested novel sensitivities based on TMM.

ALT and telomerase co-activation: The authors identified several cell lines as double-positive for ALT and telomerase. However, it remains unclear whether both pathways are functionally active. It is possible that markers such as C-circles or telomerase expression could be detected without genuine activation of the corresponding TMM. For example, cells could express telomerase without telomerase activity (e.g. impaired telomerase recruitment at telomeres), or produce low levels of C-circles without ALT serving as a TMM. Functional validation, beyond the detection of markers, would strengthen the conclusion that both pathways are simultaneously operative in double-positive cell lines.

(Remarks on code availability)

Version 1:

Reviewer comments:

Reviewer #1

(Remarks to the Author)

All my comments have been addressed.

(Remarks on code availability)

My comments regarding the code have been addressed.

Reviewer #3

(Remarks to the Author)

The authors addressed all my concerns.

(Remarks on code availability)

RESPONSE TO REVIEWER COMMENTS FOR NCOMMS-25-23547-T: *Large-scale drug sensitivity, gene dependency, and proteogenomic analyses of the spectrum of telomere maintenance mechanisms in cancer cells*

August 21st 2025

We thank the Reviewers for their favorable evaluation, insightful comments and constructive suggestions. We have revised the manuscript in accordance with each of the points raised. The revisions include new results from wet-lab validation experiments, new analyses and an expanded discussion.

Reviewers' comments are in blue and the Authors' responses are in black font below.

REVIEWER 1 COMMENTS

The manuscript "Large-scale drug sensitivity, gene dependency, and proteogenomic analyses of the spectrum of telomere maintenance mechanisms in cancer cells" conducts a large scale study of TMM in cancer cell lines. It provides a detailed atlas of proteomic and transcriptomic profiles associated with telomerase activation, and a wealth of information on TMM related assays including telomere length, TRF, TRAP, APB, and C-circle assays. The technical aspects alone will provide the field with much needed information on how the different assays compare to each other and which combination provide the potential to be a gold standard for TMM classification.

The study will provide a solid platform for the development of novel diagnostic and prognostic approaches in molecular oncology. Beyond that, important insights into TMM double positive cell lines and atypical low-ALT features cell lines are characterised, which helps to solve open questions on the interchangeability of TMMs, and thus, potential resistance mechanisms against TMM-targeting therapy.

Finally, novel insights into the sensitivity for molecular intervention are provided on the level of gene essentiality, as well as, drug sensitivity. Overall this study provides a significant advancement of the field.

Nonetheless, I remain with following comments:

Major

REVIEWER 1, Point 1

1. A high level comparison on how the defined TMM classes are distributed by cancer-of-origin for cancer-derived cell lines would be of interest. For instance, are ALT-low cases concentrated in particular cancer types? How do the ratio of these classes differ between, for instance, adenocarcinomas and sarcomas?

Authors' response

To the data indicating the distribution of TMM classes and TMM metrics across 28 different tissue types (Figure 3B of the original manuscript), we have now added data showing TMM distribution across cancer types, as follows:

- Cancer type, cancer category and tissue of origin have been added to Table S2 so that these annotations are cross-referenced to TMM measurements. This includes

morphologic cancer categories, such as adenocarcinoma, not previously indicated in this manuscript or other large cell line datasets hosted by the Wellcome Sanger Institute or Broad Institute. The morphologic cancer classifications are according to NCI Thesaurus Neoplasms by Morphology (v25.06E).

- Included an additional subpanel in Figure 3 (Figure 3C) to visually demonstrate the distribution of different TMM categories across cancer categories, such as adenocarcinomas and mesenchymal cancers.
- These findings are mentioned on page 6 of the revised manuscript as follows:
“*ALT and ALT-Low cell lines were disproportionately represented among mesenchymal cancers (soft-tissue and bone-derived sarcomas) relative to adenocarcinomas and other cancer types. ALT cell lines were also identified in cancer subtypes for which no ALT cell lines were previously known, specifically acute myeloid leukemia, gastric carcinoma, alveolar soft-part sarcoma, and chordoma (Figures 3B-C, Table S2).*”
- We have also amended the Figure 3 legend to reflect these changes.

REVIEWER 1, Point 2

2. While the TERT promoter mutations are mostly referenced by their coordinates according to the GRCh37 genome assembly in the literature, this study uses the more recent GRCh38 coordinates. It may be helpful to provide a mapping between these two nomenclatures as courtesy to the reader.

Authors' response

TERT promoter mutation coordinates from GRCh37 (gh19) and GRCh38 were both provided in Table S3 of the original manuscript in columns AM and AN. However, only GRCh38 references were provided in Figures 4B, F, G, and H.

To address the reviewer's comments, we have revised Figure 4 as follows:

- 1) Added coordinates from GRCh37 to Figure 4B so that both genome references are indicated
- 2) Indicated that the coordinates refer to GRCh38 in Figures 4F, 4G and 4H.

REVIEWER 1, Point 3

3. The training datasets for the machine learning (ML) models have a strong class imbalance, with a more than 10-fold overrepresentation of the non-ALT, respectively, TEL cases. How does the applied ML model account for that? Was downsampling of the major class considered during the training?

Authors' response

Two approaches were taken to account for the imbalance between ALT and non-ALT cell lines in ML. Firstly, down-sampling was performed by excluding cancer types that had no ALT or ALT-Low cell lines from the training set. The down-sampling step increased the representation of ALT and ALT-Low groups from 2.5% across the entire dataset to 11% and 9% for ML models, and eliminated tissue types with extreme imbalance. Secondly, the class weight option from the scikit-learn (v1.1.2) package was applied in the logistic model, where: “`LogisticRegression(class_weight='balanced', random_state=seed)`”. This function

reduces bias towards the majority category by assigning higher weights to the minority class.

We have made revisions to the Results, Methods and Figure S5A legend to clarify these steps, and mentioned the imbalance problem in the Discussion as follows:

Page 7 (Results):

“To mitigate inherent imbalance in tissue types among different TMM groups, the training model for the ALT predictor was down-sampled by restricting it to cancer types where ALT and/or ALT-Low were represented.”

Page 31, second paragraph (Methods):

“To generate ALT-specific classifiers, the training sample set was down-sampled by excluding cancer types with no ALT or ALT-Low cell lines. DP cell lines were excluded, and DN and TEL samples were combined as a non-ALT group for binary analysis.”

Page 31, third paragraph (Methods):

The multinomial logistic regression method of scikit-learn (v1.1.2) package was used for the ALT prediction models¹²⁰ using the class-weight option to mitigate bias toward the majority category (non-ALT), where: “LogisticRegression(class_weight='balanced', random_state=seed)”.

Figure S5A Legend: “To improve the balance of categories for derivation of ALT predictors, the cell line cohort was down-sampled by excluding cancer types without ALT or ALT-Low cell lines.”

Page 12, second paragraph (Discussion):

“Because the cell line panel utilized in this study was based on the availability of multi-omic and other external datasets, it is not balanced for tissue or cancer types across TMM categories. This necessitated the implementation of various strategies to control for tissue-specific effects in downstream analysis.”

REVIEWER 1, Point 4

4. A closer look at the poor performance of 3-way model on the Cell line & hMSC RNA test dataset may be instructive to improve the performance in future. Not surprisingly, the - in training overrepresented - TEL class performs best (nearly flawlessly). But how do the other classes fail in detail. Are rather DN and ALT confused with each other, or are they misclassified as TEL? By providing a 3x3 confusion matrix that provides details on which and how many cases are classified as which other class that can be easily clarified.

Authors' response

In comparison to the strong performance of the ALT (two-way) predictor, the three-way TMM classifier for ALT, TEL and DN showed modest performance on external datasets. The errors in 3-way prediction were due to confusion between DN and both ALT and TEL. Multiple factors would have contributed to this outcome, including

- (i) heterogeneity among DN cell lines, as some had a limited replicative lifespan, while others upregulated a TMM, and others sustained long-term proliferation in the absence of a TMM.
- (ii) the limited number of DN cell lines in the external cohorts – the liposarcoma cohort had no DN samples, and

- (iii) the DN cell lines in our training set were all cancer-derived and represented a variety of tissue types, whereas the DN samples in the external test set were a single cell type - mesenchymal stem cells.

Considering the limitations of the external data sets and the modest results in validation tests, we recognise that the 3-way model requires further testing, which awaits the availability of more appropriate validation cohorts, and have therefore removed the 3-way TMM analyses from the revised manuscript (Figure S5).

All references to the 3-way TMM model and classifiers have been removed from the manuscript: page 3 (Introduction), pages 7-8 (Results), 14 (Discussion), and 31 (Methods), Figure S5A and Table S4.

REVIEWER 1, Point 5

5. Among the 5 most relevant RNA features USP1 is showing up in the combined as well as the RNA model. It may be of relevance to mention that USP1 is implied in Fanconi Anemia, which creates a direct link to ALT. It would also be of interest how it univariately relates to the TMM classes on protein level.

Authors' response

In univariate analysis, USP1 RNA was significantly higher in ALT cell lines compared with all other cell lines, as shown in the adjacent graph. USP1 protein was not identified in the mass spectrometry data. We have elaborated on USP1 expression and function in the Results, page 7 of the revised manuscript, as follows:

“Notably, the top RNA features for ALT prediction were TERT (negative association) and USP1 (positive association). USP1 is a deubiquitinase that may be functionally linked to ALT through its role in DNA replication and repair involving the regulation of Fanconi Anemia proteins and PCNA^{19, 20}. In univariate analysis, USP1 RNA was significantly higher in ALT (including ALT-Low) cell lines compared with all others (P = 0.0011, two-sided Wilcoxon-ranked sum test). TERT RNA was not expressed in most ALT cell lines (Figure 4E). Neither TERT nor USP1 were detectable by DIA-MS proteomic analysis.”

REVIEWER 1, Minor comment

Minor comment:

. Figure S8 is missing the (A) label in the caption

Authors' response

We have corrected this error by adding (A) to the caption for Figure S8.

REVIEWER 1, Final comment

The manuscript should be published in any case, but may be refined by addressing these comments.

REVIEWER 1, Comments on code

The code uses well established tools and should be easily adaptable to any modern computational environment. The comments provided in the code are minimal, therefore it may require an average skill level in both of the applied programming languages (R and python) to understand it.

Minor comment: Some of the code still contains hard coded pathes to the working environment of the original programmer. Optimally, these would be substituted by neutral placeholder, maybe with a comment how to setup the project on a new computer. This would include also include which directory structure is expected by the program.

Authors' response

We have revised the code instructions, including amended paths and READ ME instructions. See Folder "Codes_V3_20250728".

REVIEWER 2 COMMENTS

Using a highly powered multiomics approach, the study presents several new aspects of telomere maintenance, mostly about ALT, including putative new gene targets (SIVA1, CHTF18, and DSCC1, TERF2IP), pathways (NEDDylation, TGF) and drug pathways (NEDD inhibitors). The study reveals further aspects of ALT, including evidence that cGAS-STING are active in ALT cells and that these cells exhibit potentially constitutive inflammatory pathway signaling. By applying phenotypic analysis with the multiomic profiling of tumor cell lines, several new ALT cell lines are identified, and others are re-categorized. The authors provide salient evidence of intra-tumoral and cell-type heterogeneity of TMMs, with telomerase and ALT co-existing as double positive cellular populations. The authors also extend prior efforts (Nonneville et al.) to develop TMM classifiers, once again, focusing primarily on ALT genomic and phenotypic markers (ATRX, DAXX, CCs, FISH). Here, machine learning of proteomic/RNA data sets are used to derived TMM predictors. Overall, this work will be a useful telomere cancer genomics resource. That being said, there is a major limitation in the absence of functional studies. These would have been very important to substantiate the work and provide new advances. Thus, the studies conclusions feel somewhat underwhelming despite the very positive outcomes mentioned above. This is the primary shortcoming identified by this reviewer.

Authors' response

This manuscript and the accompanying datasets are intended for publication as a resource to expand opportunities for studying telomere biology. The datasets generated cover the largest set of cancer models characterised for TMM to date, providing quantitative data on multiple measures of telomere biology across the entire cell line set, plus additional validation data from orthogonal assays on a substantial representative subset. The bioinformatic analyses, biomarker discovery, gene dependency and drug sensitivity analyses were performed to demonstrate the potential utility of the dataset, and to provide a platform for further investigations into telomere biology. Findings from the drug sensitivity analyses were validated by cross-referencing results from our analysis of data generated by the Wellcome Sanger Institute to data independently generated by the Broad Institute (CTRP and PRISM datasets) – see **Figure S7C** and **Table S7**.

Considering the breadth and depth of TMM analyses, we did not consider further functional validation of downstream analyses within the scope of this study. Nevertheless, in response to the reviewer's comments, we have completed wet-lab proof-of-principle experiments to validate gene dependency results for (**Figure S6**). SAMHD1 was chosen for these experiments because it was highly ranked among novel genes not previously associated with ALT. The results exemplify the potential for our results to be taken forward in further investigations of telomere biology and therapeutic strategies.

The SAMHD1 dependency validation experiments were conducted using siRNA knockdown and a panel of eight cell lines (four TEL and four ALT). SAMHD1 knockdown was confirmed by Western blot analysis, and proliferation was monitored over seven days. Cell viability was quantified in an independent assay 72 hours after siRNA transfection.

The results from these experiments are described on page 9 as follows:

“For experimental validation of dependencies associated with ALT, we focused on SAMHD1, a protein with dual functions relevant to telomere biology that was highly ranked

in differential comparisons and was not previously identified as an ALT vulnerability³⁰. Here, SAMHD1 expression was suppressed in a panel of eight cell lines (four ALT and four TEL) using two different siRNAs and a non-targeting control (Figure S6A). Proliferation was monitored over seven days, and cell viability was assessed 72 hours after transfection. Consistent with the high-throughput whole-genome CRISPR screen, SAMHD1 suppression had no impact on the proliferation of any of the four TEL cell lines. In contrast, two ALT cell lines, CAL-72 and Hs746T, which had strong dependency scores in the CRISPR screen, exhibited impaired proliferation (Figure S6B-C). The two other ALT cell lines, SAOS-2 and HuO9, showed no response to SAMHD1 knockdown, which is also consistent with the CRISPR screen. The results from the viability assay were consistent with the proliferation assay and CRISPR screen data (Figure S6C-E). Together, these results validate SAMHD1 as an ALT-associated PEG relative to TEL cells, although our data show it is not universally essential across all ALT cell lines”.

We elaborated on SAMHD1 as an ALT dependency gene in the Discussion on page 14:

“SAMHD1 is a dNTP triphosphohydrolase that regulates the availability of dNTPs for reverse transcription and replication, which was recently shown to impact telomerase-mediated telomere lengthening⁷⁶. However, the dependency of ALT cells on this gene may also be linked to SAMHD1’s dNTPase-independent role in DNA end resection during homologous recombination-mediated repair^{30, 77}.”

The experimental procedures are described in the Methods on pages 27 (Western blot analysis) and 30 (siRNA transfections, proliferation and viability assays).

REVIEWER 3 COMMENTS

This MS represents an impressive amount of work. The authors provide a comprehensive resource for the telomere biology community by analyzing over 900 cancer cell lines. They classify cell lines based on their telomere maintenance mechanism using telomerase activity, C-circle assay, and telomere content. For a subset of cell lines, they also performed telomere restriction fragment analysis and cytological evaluation of ALT features. Finally, they performed extensive proteomic analysis and determined gene dependency and drug sensitivity based on available resources. The work revealed that telomere maintenance mechanisms (TMMs) in cancer are more complex than the classical view of telomerase vs. ALT. They identified double-positive (ALT+telomerase), double-negative, and mixed TMM states. They developed machine learning-based classifiers to predict TMM status and telomerase activity from 'omic data, uncovered molecular vulnerabilities (particularly in ALT cells), and associated drug sensitivities, notably identifying ALT sensitivity to FGFR inhibition and telomerase activity correlation with Pevonedistat sensitivity. Their findings provide a new resource and suggest opportunities for TMM-targeted therapies and diagnostics.

The major strength of this study is the comprehensive approach that generated what will be an invaluable resource for the telomere biology community, with a detailed analysis of 976 cell lines, integrating telomere biology with multi-omic, CRISPR, and drug sensitivity data. Furthermore, the MS provides novel insight into gene dependencies of cell lines based on the TMM and challenges the notion of a binary choice in cancer cells between telomerase and ALT positive cells.

However, there are a few weaknesses that should be addressed and/or discussed. Potential bias in cell line selection: Certain cancer types are overrepresented or underrepresented, affecting generalizability. This is particularly important for gene essentiality and drug sensitivity. While this is an intrinsic problem with this type of approach, it should be discussed more clearly in the manuscript.

Potential for confounders: Tissue-specific effects may complicate interpretation of vulnerabilities or drug responses despite attempts to control for them.

Machine learning classifiers: While this is very interesting and has clear potential therapeutic value, it is not discussed nor verified sufficiently. Classifiers depend heavily on training datasets; applicability across broader sample types (e.g., patient biopsies) needs further testing. Additionally, there appears to be a potential contradiction between Figure 5A and Figures 6A-B, as the features that appear to be the best predictors in Figure 5A do not seem selectively enriched in the biological pathways shown in 6A-B. The authors should clarify this point.

Functional validation missing: Predictions (especially from gene dependency screens) need experimental validation to confirm therapeutic relevance. While I appreciate that the authors already provided an enormous amount of data, the lack of any type of validation limits confidence in some of the suggested novel sensitivities based on TMM.

ALT and telomerase co-activation: The authors identified several cell lines as double-positive for ALT and telomerase. However, it remains unclear whether both pathways are functionally active. It is possible that markers such as C-circles or telomerase expression could be detected without genuine activation of the corresponding TMM. For example, cells could express telomerase without telomerase activity (e.g. impaired telomerase recruitment at telomeres), or produce low levels of C-circles without ALT serving as a TMM. Functional validation, beyond the detection of markers, would strengthen the conclusion that both pathways are simultaneously operative in double-positive cell lines.

REVIEWER 3, "Potential bias in cell line selection: Certain cancer types are overrepresented or underrepresented, affecting generalizability. This is particularly

important for gene essentiality and drug sensitivity. While this is an intrinsic problem with this type of approach, it should be discussed more clearly in the manuscript.”

Authors' response

The majority of the cell lines in the pan-cancer panel used in this study were from the foundational set of the Cell Models Passport and the Sanger's GDSC project (Ref van der Meer 2018). This cell line panel was chosen because it covers a broad range of cancer types and is richly annotated with multi-omic, pharmacogenomic and gene dependency data. An additional 27 cell lines were added to the panel to enhance representation of rarer TMM types (**Tables S1 and S2**). Collectively, the cell lines were derived from 28 distinct tissues and over 60 different cancer types, including the 20 most common adult cancers worldwide, as well as representative pediatric cancers and rare cancers, such as neuroblastoma and sarcoma. We acknowledge that there may be bias in the representation of cancer subtypes due to the technical limitations of cell culture and the selection of cell lines for their tractability in wet-lab research. This limitation was stated in relation to the potential under-representation of ALT in the original manuscript (Discussion, page 12):

“ALT cell lines may be underrepresented in this panel and more broadly among cell lines because they tend to be slow-growing and may be less likely to establish in standard culture conditions than TEL cells.”

To further address this point and the Reviewer's comments, the revised manuscript includes:

- A new figure illustrating the distribution of TMM across different cancer types (**Figure S3C**) and relevant amendments to the text as described under **Reviewer 1, point 1**.
- Additional details of cancer type representation, including morphologic subtypes (**Tables S1 and S2**).
- Clarification of the number of cancer types covered by the study in the Introduction, page 3:
“...a large-scale pan-cancer telomere biology dataset covering 976 cancer cell lines derived from 28 different tissue types, and representing more than 60 cancer types.”

We have also acknowledged the complications that unbalanced cancer-type representation caused in downstream analysis – See response to Reviewer 1, point 3, including the addition to Discussion, page 12:

“Because the cell line panel utilized in this study was based on the availability of multi-omic and other external datasets, it is not balanced for tissue or cancer types across TMM categories. This necessitated the implementation of various strategies to control for tissue-specific effects in downstream analysis.”

REVIEWER 3, “Potential for confounders: Tissue-specific effects may complicate interpretation of vulnerabilities or drug responses despite attempts to control for them.”

Authors' response

We have addressed potential tissue-specific effects in downstream analysis by applying tissue-type as a covariate in differential expression, logistic regression, dependency and

drug sensitivity analyses. This is described in the revised manuscript in Results (page 8) and Methods (page 31).

Other approaches undertaken to address class imbalance are described in the response to **Reviewer 1, point 3**. Tissue-level results from drug sensitivity analyses are provided in Figures 8 and S8.

REVIEWER 3, “Machine learning classifiers: While this is very interesting and has clear potential therapeutic value, it is not discussed nor verified sufficiently. Classifiers depend heavily on training datasets; applicability across broader sample types (e.g., patient biopsies) needs further testing. Additionally, there appears to be a potential contradiction between Figure 5A and Figures 6A-B, as the features that appear to be the best predictors in Figure 5A do not seem selectively enriched in the biological pathways shown in 6A-B. The authors should clarify this point.”

Authors’ response

At the time of our study, the availability of external data sets that included both TMM and protein or gene expression data was limited to the two studies employed herein; one dataset is from a study of mesenchymal stem cells (MSCs) and cancer cell lines, and the other from analysis of liposarcoma tissue samples. We agree that further testing against patient samples annotated with both TMM data and protein or RNA expression data will be valuable. We have added this point to the Discussion on page 14, as follows;

“New datasets that include TMM, transcriptomic and/or proteomic data will be valuable for further testing these classifiers, including their capacity to predict ALT and TA from patient samples.”

Please also see our response to **Reviewer 1, point 4**, addressing the testing of classifiers and revisions.

The apparent inconsistencies between machine learning feature selection and the pathways derived from differential comparison of ALT and TEL are due to the inherent differences in the underlying analytic methods. The machine learning methodology uses sequential forward selection to build classifiers comprised of a minimum set of features that synergise in predictions. This process excludes redundant features, such as those that correlate with each other in a single pathway, and is not necessarily expected to reflect the top pathways identified by differential expression.

REVIEWER 3, “Functional validation missing: Predictions (especially from gene dependency screens) need experimental validation to confirm therapeutic relevance. While I appreciate that the authors already provided an enormous amount of data, the lack of any type of validation limits confidence in some of the suggested novel sensitivities based on TMM.”

Authors’ response

Please see our response to **Reviewer 2**.

REVIEWER 3, “ALT and Telomerase co-activation: ALT and telomerase co-activation: The authors identified several cell lines as double-positive for ALT and telomerase.

However, it remains unclear whether both pathways are functionally active. It is possible that markers such as C-circles or telomerase expression could be detected without genuine activation of the corresponding TMM. For example, cells could express telomerase without telomerase activity (e.g. impaired telomerase recruitment at telomeres) or produce low levels of C-circles without ALT serving as a TMM. Functional validation, beyond the detection of markers, would strengthen the conclusion that both pathways are simultaneously operative in double-positive cell lines.”

Authors’ response

Our results from single-cell cloning show that 19 subclones from the SCH cell line universally retained co-expression of ALT and TEL (Figure 2A), supporting the notion that both ALT and TEL contribute to the sustained survival and proliferation of these cells *in vitro*. However, we acknowledge that we currently don’t have evidence that this reflects simultaneous telomere lengthening activity of ALT and TEL in individual cells and further studies are needed (see top of page 13). Further mechanistic studies that address the Reviewer’s question will require detailed single-cell molecular analyses that are beyond the scope of this Resource paper. This study is the first to identify cancer cell lines with co-activation of ALT and telomerase, providing a valuable opportunity for mechanistic studies of the function of ALT and telomerase in this context.

We have amended the Discussion to acknowledge the need for study of TMM-double-positive cell lines on page 13 of the revised manuscript, as follows;

“Further analysis of TMM-DP cell lines may provide insight into their biology, including the role of ALT and TEL in individual cells from TMM-Dual cell lines, and the response of TMM-DP cancers to TMM-targeted therapeutics.”

ADDITIONAL AMENDMENTS

Some minor typos have been corrected, without changing the meaning.

Authorship

- An additional co-author has been added, Ran Xu from Children's Medical Research Institute. Dr Xu made a substantial contribution to the experimental work required to generate the new data shown in Figure S6.
- Added second middle initial for co-author Jennifer Koh.

Results, page 9

The following change was made to improve the flow and context of the next paragraph describing new experiments and results on ALT gene dependencies.

“Pathway analysis of TEL-associated PEGs identified strand invasion processes involving XRCC2 and RAD51D (Figures 7A and C).”

Changed to

“Conversely, these analyses revealed that ALT cells were comparatively tolerant to perturbation of genes involved in strand invasion processes (XRCC2 and RAD51D) (Figures 7A and C).”

Results and Figure legends

- Clarified statistical tests as two-sided Wilcoxon rank sum where appropriate.

Methods, page 25

- Added a statement on Ethics and Inclusion
- Under the heading “Cell Culture”, the method for single-cell sorting by flow cytometry was added, as it was overlooked in the preparation of the previous version of the manuscript.
- Added a “Code availability” statement

Acknowledgements, page 16

- Added acknowledgement of Sally Morrison and the Pinnacle Foundation for funding contributions

Supplementary Information

- Moved figures from the original Figure S6 to the revised Figure S5 to enable incorporation of new results as Figure S6; Revision of the section on machine learning, which included removal of results from the 3-way model (Figure S5), provided space for the new results on SAMHD1 gene dependency. The original Figure S6 figures were moved to the revised Figure S5 (panels B-E), and SAMHD1 figures were displayed as the revised Figure S6.
- Added a panel to Figure S2 that illustrates the gating strategy for single-cell flow sorting, as required for the editorial policy checklist. This is shown as Figure S2A.

- Added cancer subtypes and cancer categories to Table S1 for consistency with the revised Table S2.